

# Dark ice dynamics of the south-west Greenland Ice Sheet

Andrew J. Tedstone[1], Jonathan L. Bamber[1], Joseph M. Cook[2], Christopher J. Williamson[1], Xavier Fettweis[3], Andrew J. Hodson[2], and Martyn Tranter[1]

[1]Bristol Glaciology Centre, School of Geographical Sciences, University of Bristol, Bristol, UK
[2]Department of Geography, University of Sheffield, Winter Street, Sheffield, UK
[3]Laboratory of Climatology, Department of Geography, University of Liège, Liège, Belgium

*Correspondence to:* Andrew Tedstone (a.j.tedstone@bristol.ac.uk)

**Abstract.** Runoff from the Greenland Ice Sheet (GrIS) has increased in recent years due largely to declining albedo and enhanced surface melting. Some of the largest declines in GrIS albedo have occurred in the ablation zone of the south-west sector and are associated with the development of 'dark' ice surfaces. Field observations at local scales reveal that a variety of light-absorbing impurities (LAIs) can be present on the surface, ranging from inorganic particulates, to cryoconite materials

and ice algae. Meanwhile, satellite observations show that the areal extent of dark ice has varied significantly between recent successive melt seasons. However, the processes that drive such large inter-annual variability in dark ice extent remain essentially unconstrained. At present we are therefore unable to project how the albedo of bare-ice sectors of the GrIS will evolve, causing uncertainty in the projected sea level contribution from the GrIS over the coming decades.

Here we use MODIS satellite imagery to examine dark ice dynamics on the south-west GrIS each year from 2000 to 2016.

We quantify dark ice in terms of its annual extent, duration, intensity and timing of first appearance. Not only does dark ice extent vary significantly between years, but so too does its duration (from 0 % to > 80 % of June-July-August, JJA), intensity and the timing of its first appearance. Comparison of dark ice dynamics with potential meteorological drivers from the regional climate model MAR reveals that the JJA sensible heat flux, the number of positive minimum-air-temperature days and the timing of bare ice appearance are significant inter-annual synoptic controls.

We use these findings to identify the surface processes which are most likely to explain recent dark ice dynamics. We suggest that whilst the spatial distribution of dark ice is best explained by outcropping of particulates from ablating ice, these particulates alone do not drive dark ice dynamics. Instead, they may enable the growth of pigmented ice algal assemblages which cause visible surface darkening, but only when the climatological pre-requisites of liquid meltwater presence and sufficient photosynthetically-active radiation fluxes are met. Further field studies are required to fully constrain the processes by which

ice algae growth proceeds and the apparent dependency of algae growth on melt-out particulates.

## 1  Introduction

Overall mass losses from the Greenland Ice Sheet (GrIS) have increased substantially since the early 1990s (Rignot and Kanagaratnam, 2006; Rignot et al., 2011; Shepherd et al., 2012). The average rate of mass loss increased from 34 Gt yr$^{-1}$ during 1992–2001 to 215 Gt yr$^{-1}$ during 2002–2011 (Sasgen et al., 2012). During 1991-2015 the GrIS lost mass at a rate





equivalent to approximately $0.47 \pm 0.23$ mm yr$^{-1}$ of sea level rise, with a peak contribution in 2012 of 1.2 mm (van den Broeke et al., 2016). Increases in mass losses since 2009 have been dominated by increased surface runoff, with only 32 % of the total loss in this period attributable to solid ice discharge (Enderlin et al., 2014). It is therefore essential to understand the processes which control surface melting in order to be able to quantify the contribution of the GrIS to sea level rise over the

coming century.

Surface melting is controlled primarily by albedo. A lower albedo permits more absorption of shortwave radiation, which in turn leads to enhanced ice melting, and so albedo is the dominant factor governing surface melt variability in the ablation area (Box et al., 2012). The effective albedo of the GrIS is controlled by external factors including solar zenith angle, atmospheric composition and cloud cover, as well as the inherent optical properties of the surface. For both snow-covered and bare-ice

surfaces these inherent optical properties are modified by (a) ice grain metamorphism, (b) meltwater on the surface or in interstitial pores, and (c) light-absorbing impurities (LAIs) including biological and mineralogical substances (Gardner and Sharp, 2010).

Declines in GrIS bare-ice albedo have an immediate impact on runoff from the GrIS and hence the surface mass balance (SMB). Decreases in the SMB since 1991 are predominantly due to enhanced runoff from bare-ice, low-lying (<2000 m.a.s.l.)

parts of the ice sheet (van den Broeke et al., 2016). Since around 2000 the surface albedo of several sectors of the GrIS has often been significantly lower each summer than was observed during the 1990s (He et al., 2013). GrIS summer albedo showed a negative trend during 2000-2012, with the largest decreases observed in western Greenland (Stroeve et al., 2013). Some of the decline in albedo can be attributed to increases in bare-ice extent. The GrIS-wide bare-ice ablation zone extent increased by 4.4 % per year on average from 2000 to 2014, although with substantial inter-annual variability of between 5 % and 16 %

of the ice sheet (Shimada et al., 2016).

In the ablation zone of the south-west GrIS, albedo lowered by as much as 18% from 2000 to 2011 (Box et al., 2012). The south-west has seen the greatest increase in bare-ice extent, by on average 5.8 % per year, with a mean extent of 56,603 km$^2$ during 2000–2014 (Shimada et al., 2016). However, increasing bare-ice extent alone is insufficient to explain the declining albedo. Remotely-sensed optical imagery for this sector shows a band of relatively darker ice within the bare-ice ablation zone

which recurred annually in the same location over the period 2001–2007, beginning 20–30 km inland from the ice sheet margin and extending up to ∼50 km wide, which has been postulated to be caused by LAIs (Wientjes and Oerlemans, 2010). LAIs on snow/ice surfaces reduce reflectance the most in the visible part of the solar spectrum (Warren, 1984; Painter et al., 2001; Bøggild et al., 2010), and this effect enabled Shimada et al. (2016) to quantify the inter-annual extent of dark ice — both GrIS-wide and for the south-west sector — by applying an empirically-derived reflectance threshold to the 620-670 nm band

of MODIS satellite imagery acquired in July each year. They found that dark ice extent varied substantially between years, both GrIS-wide (from 3575 to 26,975 km$^2$) and in the south-west (from 575 to 15,025 km$^2$).

There are a range of possible causes of dark ice on the GrIS. One is the melt-out of particulates from ablating ice. Wientjes et al. (2012) acquired shallow ice cores from the south-west sector in which they found dust that they dated to the Late Holocene. They therefore suggested that the dust was deposited in the accumulation zone and flowed with the ice down to

the ablation zone where it has been melting out in recent years, causing darkening of the surface. However, they were not

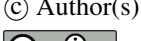



able to measure absolute concentrations of dust in their ice cores to compare to non-dark regions of the ice sheet. Meanwhile, Shimada et al. (2016) found a statistically significant correlation (0.69) between July dark ice extent and air temperature (and hence surface melt rates, potentially causing enhanced particulate melt-out) in the south-west sector, but did not identify the responsible component of the surface energy balance.

Another potential source of darkening is the deposition of black carbon and other inorganic impurities by wet and dry atmospheric deposition, which has been investigated in ice and snow elsewhere (Warren and Wiscombe, 1980; Warren, 1984; Warren and Wiscombe, 1985; Gardner and Sharp, 2010). However, black carbon appears unlikely to explain variations in dark ice on the south-west GrIS. First, concentrations of black carbon in snowpack in the north-western snow sector are too low to cause any appreciable darkening and have been stable or even slightly declining over the past decade (Polashenski et al., 2015).

Second, fire events in North America and Eurasia became rarer from 2002 to 2012 (Tedesco et al., 2016). Third, there is no recent statistically significant trend in aerosol flux deposition estimates along the south-west margin of the ice sheet (Tedesco et al., 2016).

In addition to inorganic impurities alone, the ice sheet can be darkened by ice surface habitats. Cryoconite is an aggregate of inorganic materials bound together by extracellular polymers produced by microorganisms, predominantly cyanobacteria

(Wharton et al., 1985; Takeuchi et al., 2001; Hodson et al., 2008; Cook et al., 2016). Cryoconite absorbs more shortwave radiation than the surrounding ice and so, when the surface energy balance is dominated by shortwave radiation, ice overlain by cryoconite will melt more quickly than the surrounding ice. This produces water-filled cryoconite holes with a floor of biologically-active sediment (Gribbon, 1979; Cook et al., 2016). These holes range from a few centimetres to several metres in diameter and depth (MacDonell and Fitzsimons, 2008), can cover a large part of the ablation zone (Hodson et al., 2008), and

have been observed to occur in the south-west region of the GrIS (Stibal et al., 2012; Cook et al., 2012; Chandler et al., 2015; Stibal et al., 2015; Cameron et al., 2016). Hole formation increases the albedo relative to dispersed cryoconite by sequestering the low-albedo cryoconite from the ice surface at depth beneath a reflective layer of meltwater (Bøggild et al., 2010). Occasional stripping events cause redistribution of aggregates onto the ice surface and subsequent new hole formation (MacDonell and Fitzsimons, 2008; Irvine-Fynn et al., 2011).

Distinct from the assemblages of microorganisms associated with cryoconite holes, ice algae can bloom in the upper few centimetres of bare melting ice. Abundant assemblages of ice algal communities have been reported on bare ice in both west (Uetake et al., 2010; Yallop et al., 2012) and east Greenland (Lutz et al., 2014). Ice algae produce specialist pigments which absorb UV and visible wavelengths, protecting the photosynthetic apparatus from excessive radiation (Dieser et al., 2010; Yallop et al., 2012; Remias et al., 2012). These pigments may be a significant source of darkening to GrIS surface ice (Yallop

et al., 2012; Lutz et al., 2014).

The influence of cryoconite, cryoconite hole processes, and/or ice algal assemblages on the substantial inter-annual variability apparent in dark ice extent of the GrIS is currently unknown. Whilst Shimada et al. (2016) proposed cryoconite sequestration into cryoconite holes as the mechanism underlying the negative correlation (-0.52) between ice-sheet-wide July dark ice extent and shortwave radiation, this relationship did not hold when examined for the south-west sector alone. Additionally, although

opposed pyranometer measurements (300–1100 nm) demonstrated that local algal bloom patches had lower albedo at these



wavelengths than snow without visible blooms, broadband albedo measurements relevant for energy balance have not been isolated from grain evolution, meltwater ponding and abiotic impurities.

In this study we aim to identify the 'top-down' controls of significant variability in dark ice extent between successive melt seasons in the south-west of the GrIS. We first characterise the inter-annual dark ice dynamics of the south-west GrIS using visible satellite imagery to quantify dark ice in terms of its extent, duration, intensity and the timing of its appearance each year. We then examine the extent to which inter-annual variations in dark ice dynamics are controlled by prevailing seasonal meteorological and climatological conditions and how they could drive surface darkening through three potential processes: (1) inorganic particulate deposition, (2) cryoconite hole processes and (3) growth of ice algal assemblages.

## 2 Data and Methods

### 2.1 Identification of dark ice

We used the MOD09GA Daily Land Surface Reflectance Collection 6 product, which is derived from data acquired by the MODIS sensor on board NASA's Terra satellite. Collection 6 products include improved calibration algorithms to correct for MODIS sensor degradation on Terra (Lyapustin et al., 2014) which was responsible for an apparent decline in GrIS dry snow albedo over the last decade or so (Polashenski et al., 2015; Casey et al., 2017). We used the MOD10A1 Daily Snow Albedo Collection 6 product, which contains a cloud discrimination layer, to identify and discard pixels covered by cloud. Our full time series encompasses daily observations between May and September from 2000 to 2016 but here we concentrate mainly on observations made during JJA.

Both MODIS Level-2 products are delivered on a sinusoidal grid which causes significant distortion over the GrIS and prevents simple comparison with meteorological fields output by the regional climate model MAR (Sect. 2.5). We therefore first re-projected the MODIS data to the Polar Stereographic projection used by MAR using nearest-neighbour re-sampling, yielding a spatial resolution of $\sim 600 \times 600$ m.

We detected bare ice and then dark ice within bare-ice areas by applying thresholds to reflectance values ($R$) (Shimada et al., 2016). For bare ice we adopted $R_{841-876nm} < 0.6$. To detect dark bare ice we used $R_{620-670nm} < 0.45$. Pilot field spectra acquired in July 2016 indicate that this slightly higher threshold — compared with $R_{620-670nm} < 0.4$ used by Shimada et al. (2016) — captures dark ice more accurately (Appendix A). However, we note that we do not know what this dark ice threshold represents physically. For instance, we do not know the composition, concentration or sub-pixel spatial distribution of LAIs which result in $R < 0.45$, nor whether the threshold represents the minimum amount of darkening required to be detectable either in the field or remotely. Thus, our threshold may not capture all sources of darkening or precisely identify precisely the timings of dark ice dynamics. It is also noteworthy that $R_{620-670nm}$ straddles a transition zone between wavelengths mostly influenced by LAIs and wavelengths mostly influenced by grain evolution and interstitial water.



## 2.2 Selection of common area

We defined a common area of maximum dark ice extent. This enabled the spatial sampling area to be held constant when calculating inter-annual statistics. We chose this approach over defining different areas of dark ice for each year because then the spatial sampling area would have changed dramatically from one year to the next, whereas we are mainly interested in the primary drivers of inter-annual variability in dark ice dynamics.

To define the common area, first, in each year, we identified the pixels which were flagged as dark on at least 10 d during June-July-August (JJA). Then, we retained only those pixels which went dark in at least 4 y of our time series. Finally, we removed all dark pixels which occurred within ∼1 km of the ice sheet margin. The common area is depicted in Fig. 1 and covers ∼10,400 km$^2$.

## 2.3 Metrics of dark ice dynamics

We derived four metrics to characterize spatio-temporal variations in dark ice. First, annual extent ($D_E$) corresponds to all the pixels within the common area which were dark for at least 5 d in each year. Second, annual duration ($D_D$) was defined at each pixel in the common area as the percentage of all daily cloud-free observations made in each JJA period, and is thereby normalised for cloud cover. Third, intensity ($D_I$) was defined as the mean daily reflectance over 620–670 nm in the common area, and annual intensity ($\bar{D}_I$) as the mean of all days in each JJA period. A lower value of $D_I$ or $\bar{D}_I$ therefore means that dark ice intensity was greater. $D_I$ is on a continuous scale and so is independent of the stringent dark ice presence threshold defined in Sect. 2.1. Only days in which at least 50 % of the common area was cloud-free were included in the calculation. Fourth, normalised darkness was expressed as

$$D_N = \frac{D_D}{\bar{D}_I \cdot 100} \tag{1}$$

and therefore provides a combined indicator of both the duration and intensity of dark ice presence.

We note that cloud cover was present to some degree over the common area in almost every day of our time series, which prevented us from quantifying daily dark ice extent.

## 2.4 Timing of bare ice and dark ice appearance

At each pixel and for each year we identified the date on which (a) bare ice emerged from underneath the melted snowpack ($t_B$) and (b) dark ice appeared ($t_D$), if at all. In both cases we used a 7 d rolling window on the relevant time series of reflectance at each pixel. Each year, we identified the first rolling window at each pixel that contained at least 3 days of bare or dark ice (not necessarily consecutive), with any remaining days in the window only allowed to be cloudy. We then selected the first day of bare or dark ice appearance from within the chosen window. This windowing strategy enabled us to minimise the likelihood of false-positive identification of bare and dark ice appearance dates which would have occurred if only looking at daily observations in isolation and also allowed us to ameliorate for cloud cover.



Finally, we calculated the median day-of-year of bare ice appearance for each year ($\tilde{t_B}$) from the pixel-level data.

## 2.5 Meteorological and climatological data

We performed simulations of meteorological conditions over the GrIS using version 3.6.2 of Modèle Atmosphérique Régional (MAR), a regional climate model (Fettweis et al., 2017). The model was run on an equal-area 7.5 x 7.5 km resolution grid for the whole of Greenland and was forced at its boundaries every 6 h by ECMWF ERA-Interim re-analysis data. For comparison with dark ice dynamics we down-sampled the MODIS-defined common area to MAR's resolution. We calculated mean shortwave-down ($SW \downarrow \prime$), longwave-down ($LW \downarrow \prime$), and sensible heat flux ($SHF\prime$) anomalies in the common area for each JJA relative to 1981–2000. We also calculated the mean daily snow depth in the common area from April to August each year, total snowfall (from $\tilde{t_B}$ to 31 August) and total rainfall (during JJA).

We characterised near-surface air temperatures in two ways. First, we defined the mean air temperature during JJA as $T$. Second, we defined the number of days in each JJA period on which the common area's daily *minimum* near-surface air temperature exceeded 0 °C as $\sum T > 0$.

As introduced previously, cryoconite holes form and tend to be sustained under $SW \downarrow$ dominant conditions. This suggests that they are likely to melt out if the surface energy balance shifts to $LW \downarrow$ or $SHF$ dominant conditions. We therefore estimated the daily energy available to cause the melt-out of cryoconite holes using

$$SW_{net} = SW \downarrow \cdot (1 - \alpha) \tag{2}$$

$$LW_{net} = LW \downarrow - LW \uparrow \tag{3}$$

$$MOF = SHF + LW_{net} - SW_{net} \tag{4}$$

where $\alpha$ was the daily mean MOD10A1 albedo over the common area (only on days with <50 % cloud cover) and $LW \uparrow$ was 315.6 $\mathrm{Wm}^{-2}$ for melting ice surfaces as defined by Cuffey and Paterson (2010). $M\bar{O}F$ corresponds to the mean JJA $MOF$.

We used the monthly Greenland Blocking Index (GBI) (Hanna et al., 2016) to consider the role of the synoptic atmospheric circulation in dark ice dynamics. The GBI is the mean 500 hPa geopotential height for the 60–80 °N, 20–80 °W region and therefore provides a measure of the extent of high-pressure blocking over Greenland. We calculated the mean GBI for each JJA period.

We tested for relationships between metrics of dark ice dynamics and meteorology using ordinary least squares regression.

## 3 Results

### 3.1 Characteristics of dark ice dynamics

Shimada et al. (2016) identified a general trend of increasing $D_E$ over time but also saw that $D_E$ on the south-west GrIS varied dramatically between years. We found similar characteristics in our expanded time series (Fig. 1). $D_E$ ranged from almost no





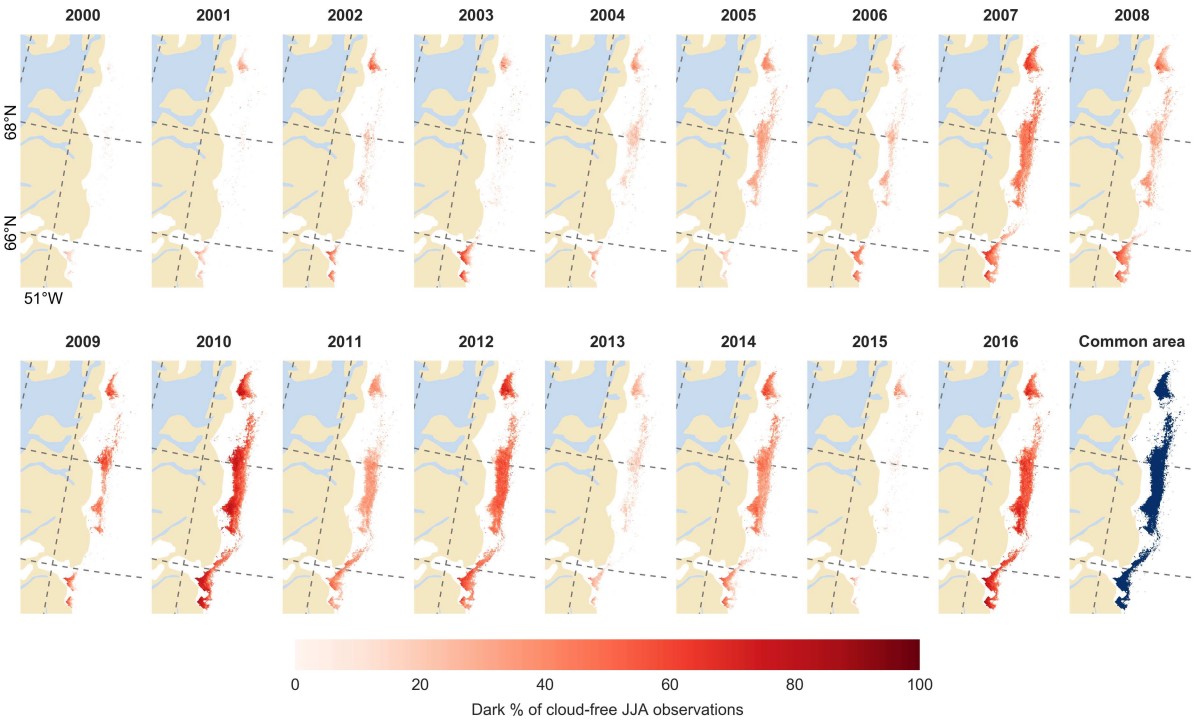

**Figure 1.** $D_E$ and $D_D$ on the south-west GrIS each summer from 2000 to 2016, expressed as a percentage of the total daily cloud-free observations made during June-July-August (JJA). In each year, pixels that are dark for fewer than 5 days are not shown. Bottom-right panel: common area of dark ice used for inter-annual comparisons.

dark ice identified (2000, 2001 and 2015), to wide, contiguous areas of dark ice stretching from 65.5 to 69°N (2007, 2010, 2011, 2012, 2014 and 2016).

In addition, there was substantial inter-annual variability in $D_D$ during JJA. Generally, when $D_E$ was high, $D_D$ was also high, especially in 2010, 2012 and 2016. Moreover, the extension of our time series to encompass June through August revealed

5  relatively large $D_E$ and $D_D$ in 2005, 2007, 2008, 2009, 2011 and 2014 which has not been captured previously.

Examination of $D_I$ (Fig. 2) shows that most dark ice presence was concentrated into the months of July and August. In some years (2010 and 2016) more significant darkening of the ice sheet surface was observed as early as mid June. In years when substantial darkening occurred there was a time lag following $\tilde{t_B}$ of ~10–15 d. $D_I$ tended to increase over the season. Variability in $D_I$ at daily to weekly timescales was minimal compared to the magnitude of variability over inter-annual

10 timescales. Dark ice usually persisted until the onset of anti-cyclonic, cloudy conditions (Fig. 2, days shaded gray) and snowfall during late August and September, which buried the bare-ice surface under snowpack for the winter period.

We did not find any evidence that the dynamics of dark ice in one year controlled dark ice dynamics the following year. There were been years of higher $D_N$ recently (2012, 2014, 2016) interspersed with years of much lower $D_N$ (2013, 2015). Moreover,





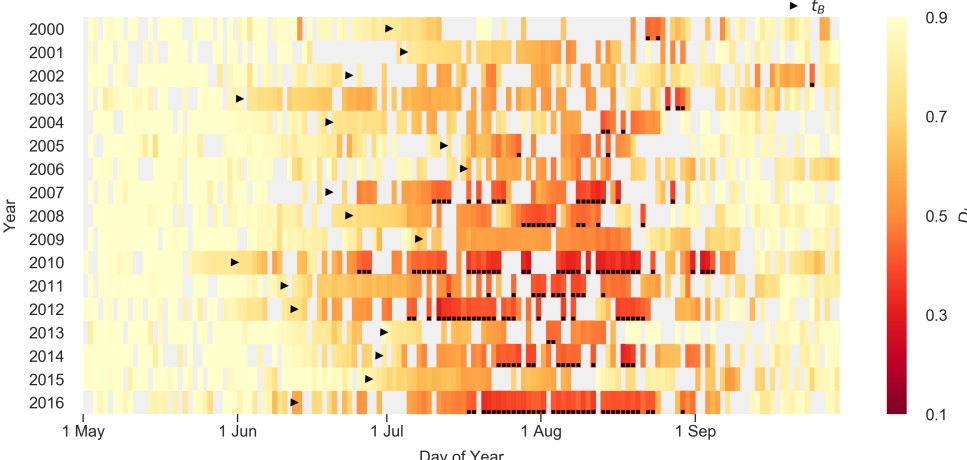

**Figure 2.** $D_I$ in the common area during May to September from 2000 to 2016. Only days on which at least 50 % of the common area is cloud-free are shown. Black squares denote days on which the entire common area had $D_I < 0.45$.

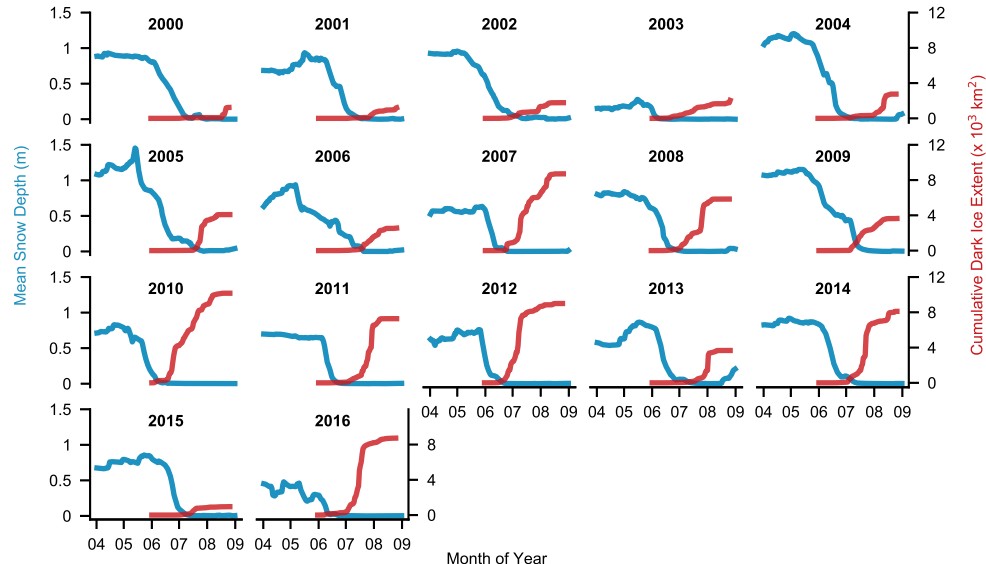

**Figure 3.** Average snow depth (blue) and cumulative $D_E$ (red) in common area during April to August each from 2000 to 2016.

$D_I$ values at end of one melt season were generally significantly different to those in the period after $\tilde{t_B}$ the following year (Fig. 2).

We used $t_D$ to calculate cumulative $D_E$ in the common area through the summer (Fig. 3, red lines). In several years $D_E$ was very small (2000, 2001, 2003, 2015). In years when medium $D_E$ occurred (e.g. 2005, 2006, 2008, 2009, 2013), dark ice appeared step-wise through July and into August. This step-wise appearance also occurred in the high $D_E$ years of 2007 and





2010. In contrast, the widespread expansion of $D_E$ in 2011, 2012, 2014 and 2016 occurred rapidly over just a few days in July. In particular, we found 28 large, single-day expansions in dark ice extent in our time series (defined as >520 km$^2$, equivalent to ~5 % of the common area). These large changes were not explicable by gaps in our time series owing to cloud cover: the median number of preceding days when cloud cover was >50 % was 0, and the mean common area covered by cloud in the

preceding 7 days was 34 %. There tended to be minimal further dark ice expansion in August. As shown by $D_E$ (Fig. 1) and $D_I$ (Fig. 2), the magnitude of dark ice tended to persist for the rest of the summer season (Figs. 1 and 2).

## 3.2 Controls on dark ice presence

In years when the ice went dark then $t_D$ and $D_D$ tended to be spatially invariant across the common area. This suggests that the driver/s of dark ice presence is/are synoptic, governing dark ice dynamics over the whole common area. We therefore used

meteorological and climatological variables representative of the common dark ice area to examine their potential impact upon dark ice dynamics.

### 3.2.1 Snow

Snow can control dark ice dynamics in at least two major ways: (a) the thickness of the snowpack from the preceding winter will, in combination with air temperatures, control $\tilde{t_B}$; and (b) snowfall which occurs during the melt season will at least

temporarily obscure the bare ice surface.

Fig. 2 shows $\tilde{t_B}$ and Fig. 3 shows the mean snowpack depth from April through to August each year. In the years of longest $D_D$ and greatest $D_I$ (2007, 2010, 2012, 2016) bare ice appeared by roughly mid June, compared to other years when bare ice did not appear until early to mid July.

Earlier $\tilde{t_B}$ is not just a function of total snowfall during the preceding winter but is also strongly dependent on the progression

of melting during spring which, in extreme cases such as 2016, began as early as April, introducing liquid water to the snowpack and accelerating its warming despite additional snowfall in May. On the other hand, in years such as 2015, significant melting did not occur for the first time until mid June. Not only was winter snowfall relatively large compared to other years in our study, but more snowfall occurred around the start of June just before the melt season started. Nevertheless, in general a thinner winter snowpack favoured earlier $\tilde{t_B}$, and earlier $\tilde{t_B}$ in turn favoured increased $D_N$ (R$^2$ 0.51, p < 0.01, Fig. 5f). Last, when

further snowfall occurred during summer (Fig. 4b) then $D_N$ tended to be lower (R$^2$ 0.36, p < 0.05).

### 3.2.2 Atmospheric energy fluxes

$SW \downarrow \prime$ was consistently positive from 2007 onwards but continued to show substantial inter-annual variability (Fig. 4a). There was no statistically significant relationship between JJA $SW \downarrow \prime$ and $D_N$. Unlike $SW \downarrow \prime$, from 2000 to 2007 $LW \downarrow$ anomalies were consistently positive and then after 2007 the sign became more variable, with both positive and negative anomalies

occurring (Fig. 4a). Like $SW \downarrow \prime$ there was no statistically significant relationship with $D_N$.

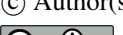


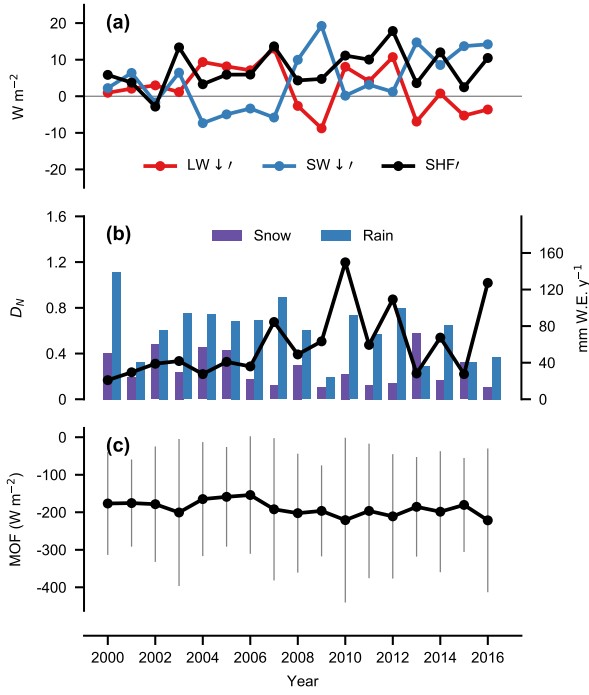

**Figure 4.** JJA meteorology and $D_N$ from 2000 to 2016. (a) $SW \downarrow \prime$, $LW \downarrow \prime$ and $SHF\prime$. (b) $D_N$ total snow inputs from date of snow clearing until 31 August and rain inputs during JJA. (c) $M\bar{O}F$, $\pm 3\sigma$.

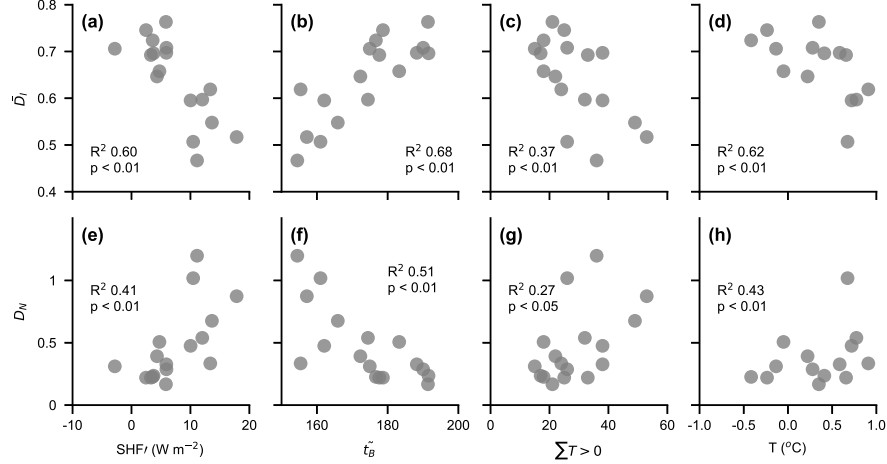

**Figure 5.** Relationships between meteorological indicators and $D_I$ (upper) and $\bar{D}_N$ (lower).





$SHF\prime$ was consistently positive throughout the time series. There was a significant positive correlation between $SHF\prime$ and $D_N$ ($R^2$ 0.41, $p < 0.01$, Fig. 5e).

We examined the likelihood for cryoconite hole melt-out (causing redistribution of cryoconite materials onto the ice sheet surface) using $M\bar{O}F$ (Fig. 4c), which describes the amount of energy available to melt cryoconite holes out of their weathering crust. Positive $M\bar{O}F$ signifies that longwave and sensible heat fluxes dominate the energy balance, which will cause spatially 'even' surface melting as opposed to spatially heterogeneous melting permitted by stronger absorption of $SW\downarrow$ where cryoconite material is present. $M\bar{O}F$ was negative in all years, and all days within $3\sigma$ of the mean were also negative, which suggests that the positive $M\bar{O}F$ conditions required for the melt-out of cryoconite holes were seldom (if ever) met.

As liquid meltwater constitutes a pre-requisite for algal growth, we assessed the likelihood of continuous liquid meltwater presence on the ice surface over each 24 h cycle using $\sum T > 0$. We found a positive correlation between $\sum T > 0$ and $\bar{D}_I$ ($R^2$ 0.37, $p < 0.05$, Fig. 5c) and to a lesser extent with $D_N$ ($R^2$ 0.27, $p < 0.05$, Fig. 5g). Greater $\sum T > 0$ coincided with higher $SHF$ ($R^2$ 0.55, $p < 0.01$). Moreover, we found that single days of large dark ice area expansion were associated with a median of 3 days of continuous (24 h) melting, compared to 0 days for the rest of the time series. These sudden increases in dark ice extent were associated with higher absolute sensible heat fluxes, with a mean of $57\pm24$ $\mathrm{Wm^{-2}}$, equivalent to 96% more sensible heat than on the days from the start of dark ice expansion until a maximum $D_E$ of 90 % of the common area.

Last, we examined whether the dark ice dynamics have any relationship with the GBI. We found a positive correlation between the JJA GBI and $D_N$ ($R^2$ 0.46, $p < 0.05$).

### 3.2.3 Rainfall

Rainfall can occur in the ablation zone of the GrIS during summer. Limited observations from elsewhere in the cryosphere indicate that whilst the direct melt impact of rainfall upon melt rates is generally limited, rain can affect melt indirectly by increasing the liquid water content of the ice surface, reducing its albedo (Hock, 2005). Total JJA rainfall in this sector ranged from 30 mm w.e. to as much as 140 mm w.e. (Fig. 4b). However, there was no statistically significant relationship between JJA total rainfall and $D_N$.

Eyewitnesses on the ice sheet surface have observed that the weathering crust can be stripped back to high-density bare ice during rainfall events (potentially dispersing cryoconite material) but that it tends to reform within days. We therefore also examined the impact of each JJA rainfall event upon $D_I$. We selected all rainfall (and snowfall) events of >1 mm W.E. d$^{-1}$ across our common area in our time series. Then, we calculated the change in $D_I$ using the closest observations immediately before and after the rainfall event. We found no systematic change in $D_I$ caused by rainfall events: in some cases $D_I$ increased while in other cases it decreased. This was the case whether or not mixed rainfall and snowfall events were excluded from analysis, although we note that MAR may not adequately discriminate between rainfall and snowfall over ice surfaces.



## 4    Discussion

At outlined in Sect. 1, a number of processes have been proposed to explain the dark ice dynamics on the south-west of the GrIS. Our characterisation of dark ice in terms of $D_E$, $D_D$, $D_I$ and $t_D$, when combined with analysis of the prevailing meteorological conditions estimated by MAR, allows us to consider the extent to which each proposed process fits with our observations of dark ice dynamics.

### 4.1    Variability driven by inorganic particulate deposition

There are two primary ways in which inorganic particulate matter can arrive on the ice-sheet surface: (1) by wet and dry atmospheric deposition, and (2) melt-out of material trapped in the ablating ice. Previous research indicates that there is no relationship between albedo reductions and the number of fires occurring over North America and Eurasia nor with modelled atmospheric aerosol fluxes (Tedesco et al., 2016). The only published measurements of black carbon on the GrIS are from the north-west (Aoki et al., 2014; Polashenski et al., 2015) and at only a few ppb are too low to explain the observed reduction in reflectance in the south-west.

Atmospheric deposition events would presumably have to occur in only years of high $D_N$ in order to explain the spatio-temporal patterns in dark ice that we observed. In dark years, $D_I$ increased gradually over the summer and so we postulate that deposition would need to occur over at least several days after $\tilde{t_B}$. More problematic is that maximum $D_E$ is relatively invariant and begins c. 20 km inland rather than at the margin, whereas atmospheric deposition would presumably occur over a more dispersed area.

The well-defined geometry of the dark-ice area between years lends support to the hypothesis that the dark-ice area is caused by the melt-out of particulates trapped in the ablating ice (Wientjes and Oerlemans, 2010; Wientjes et al., 2012). Warmer air temperatures in darker years ($R^2$ 0.43, $p < 0.01$; Fig. 5d/h) support the idea that more material can melt out in these years and thereby contribute to darkening, possibly acting as a positive feedback mechanism. However, our dynamics observations suggest that particulate melt-out is not the primary source of darkening. First, there is more variability in $D_E$ and $D_D$ than we would expect if total summer ablation alone determined darkening by controlling the quantity of particulates melting out. $D_E$ was negligible in several years (Fig. 1), yet melting in this area occurred in all years (e.g. van den Broeke et al., 2011; van As et al., 2016). Second, particulates are unlikely to be dispersed homogeneously through the ice column and so the concentration of melt-out particulates emerging at the surface will change non-linearly with respect to the ice-melt rate. This could explain why $D_E$ is negligible in several high-melt years. However, the wavy patterns of surface darkening at decimetre scales observed by Wientjes and Oerlemans (2010) are indicative of dispersion of previously well-defined particulate horizons by vertical shear due to ice flow, which suggests that particulates melted out in each summer of our time series. Third and most critically, in order to explain non-dark years such as 2013 and 2015, the particulate material responsible for substantial darkening during 2010–2012 and 2014 would have to be evacuated from the ice-sheet surface at the end of summer to explain both dark ice that summer and the lack of dark ice the year after. More broadly, we did not find a year-on-year increase in $D_I$ that we would expect if melt-out particulates were accumulating at the ice-sheet surface (Fig. 2). We also found that in years of high $D_N$ the





onset of high $D_I$ was delayed by $\sim$10-15 d after $\tilde{t_B}$ (Fig. 2). This may be attributable to superimposed ice formation but the prevalence of this process on the GrIS remains poorly understood (Larose et al., 2013; **?**). Nevertheless, if particulate materials were still present on the surface from the previous melt season we would expect high $D_I$ almost immediately after bare ice appearance.

Overall, our observations provide little support to the hypothesis that particulate deposition causes surface darkening. We are unable to identify a mechanism by which the ice-surface mass flux of particulate material could change over timescales commensurate with dark ice dynamics.

## 4.2   Variability driven by cryoconite hole processes

Shimada et al. (2016) hypothesised that the opposing processes of cryoconite hole formation (under $SW \downarrow$ dominant condi-
tions) versus melt-out (under $LW \downarrow$ dominant conditions) could explain inter-annual variability in $D_E$.

There are several noteworthy limitations to investigating cryoconite hole processes from satellite observations. First, cryoconite hole processes occur over decimetre scales and so we may not be able to capture their variability using 500 m MODIS imagery. Second, the reflectance of ice surfaces with cryoconite holes varies strongly as a function of viewing angle (Bøggild et al., 2010), and so observations made by MODIS — which has a 'push-broom' scanning assembly — will vary depending
on how near to nadir the angular Instantaneous Field Of View (IFOV) is. This is likely to impact the broadband albedo value from MOD10A1 used to calculate $SW_{net}$ as part of $MOF$ in a way that is not currently known.

Shimada et al. (2016) suggested that low $D_E$ in July 2011 followed by widespread $D_E$ in July 2012 could be attributable to cryoconite hole wash-out during anti-cyclonic conditions in late August and September 2011. However, our results reveal a different spatio-temporal pattern of darkening: we found that the common area did go dark in 2011, but that this did not begin
until late in July. Maximum $D_I$ was reached during August, which Shimada et al. (2016) omitted from their analysis. In 2012 we observed an early onset of high $D_E$, with rising $D_I$ as the season continued, whereas under a return to $SW \downarrow$ dominant conditions we would expect $D_I$ to gradually decrease as cryoconite hole surfaces deepen, albeit non-linearly because cryoconite holes cease deepening once they are in equilibrium with their surroundings (Gribbon, 1979). This also makes it difficult to explain low $D_E$ in 2013, as cryoconite holes would have needed to form over a short period at the end of summer 2012 in
order to sequester cryoconite particles at depth.

We also looked at intra-annual variations in dark ice dynamics when considering the potential for variability to be driven by cryoconite hole processes. Field observations of cryoconite hole morphology show that cryoconite holes form within the ice weathering crust over timescales of a few days (Cook et al., 2016). However, in dark years, $t_D$ was relatively synchronous across the common area. Importantly, $D_I$ then increased as the season continued, suggesting that episodic cryoconite hole flushing
and reformation is unlikely. Moreover, the lack of energy available for cryoconite hole melt-out over seasonal timescales (Fig. 4c) suggests that variability in the areal extent of cryoconite holes forced by changes in the dominant component(s) of the surface energy balance cannot explain dark ice dynamics. Our only evidence in support of cryoconite hole processes is that large single-day increases in dark ice extent were associated with higher absolute $SHF$, which could still cause transient



hole-flushing events during the melt season. However, the rest of our evidence strongly suggests that cryoconite hole processes are not responsible for inter-annual dark ice dynamics.

### 4.3   Variability driven by ice algal assemblages

Last, we examined evidence for the role of ice algal assemblages as the principal driver of dark ice dynamics. In addition to
typical light-harvesting pigments characteristic of green microalgae (Remias et al., 2009), ice algae produce a unique UV-VIS absorbing purpurogalin pigment that presumably affords protection from the significant radiation experienced in GrIS surface habitats (Remias et al., 2012). Given this pigmentation, ice algal blooms are known to impact visible reflectance at local (metre) scales (Yallop et al., 2012; Lutz et al., 2014). Knowledge of the regulation of temporal and spatial patterns in ice algal biomass (and thus pigmentation) in surface habitats is limited (Yallop et al., 2012; Chandler et al., 2015), but the fundamental
pre-requisites for algal life are known, including liquid water, nutrient resources and photosynthetically active radiation (PAR, 400–700 nm).

The significant positive relationship identified between $\sum T > 0$ and $D_I$ ($R^2$ 0.37, p < 0.01, Fig. 5g) supports the role of ice algae in ice sheet darkening, as do the single-day increases in $D_E$ of >5 % of the common area, which were generally preceded by several days of continuous positive air temperatures; both of these observations are indicative of liquid meltwater
presence. Ice algae require liquid water in order to grow, and ice surfaces are reservoirs of potentially viable propagules that can become active when they encounter sufficient liquid water of appropriate chemistry (Webster-Brown et al., 2015). Minimum air temperatures above 0ºC required for the presence of liquid water will facilitate growth of ice algae. As blooming progresses, the relationship between liquid water availability and algal proliferation may be strengthened by the establishment of a positive feedback loop via albedo reduction. For example, blooms of snow algae have been shown to result in surface albedo reduction,
increased heat retention at the snow surface, and thus enhanced melting and liquid water availability for continued algal growth (Lutz et al., 2016). Climatically, enhanced liquid meltwater presence in dark years — especially continuing through the night when $SW\downarrow$ tends to zero — is also partially attributable to increased $SHF\prime$, with a positive correlation observed between $SHF\prime$ and $D_N$ ($R^2$ 0.41, p < 0.01, Fig. 5e) in our study. Thus a combination of greater $\sum T > 0$ and higher $SHF\prime$ may regulate inter-annual liquid water availability in ways critical to ice algae growth, thereby governing whether or not dark ice
appears.

$\tilde{t_B}$ has a first-order impact on whether the common area is dark in any given year, with later appearance associated with lower $D_E$ (Fig. 5b,f). $\tilde{t_B}$ will significantly impact PAR availability at the ice surface. If bare ice appears in early to mid June, it will receive PAR over several complete diurnal cycles, unlike in years when bare ice does not appear until July. Although ice algae likely experience excessive irradiance over the ablation season, as evidenced by their production of 'sun-screen' pigments
(Remias et al., 2012), a minimum threshold of PAR (or photo-period duration) may be required to allow bloom initiation, which would be favoured by earlier $\tilde{t_B}$. Alternatively, variability in $\tilde{t_B}$ may impact algal blooms (and thus darkening) via the timing of nutrient inputs to surface ice, or due to the formation of superimposed ice. With delayed snow line retreat, percolating snow melt in spring/early summer may release snow pack nutrients to surface ice (Larose et al., 2013) before PAR is available to allow algal utilisation, stalling bloom formation. It may also result in sustained presence of superimposed ice (Larose et al., 2013;



Chandler et al., 2015), preventing PAR penetration to the previous year's ice algal cells and initiation of growth. However, we found no significant relationship between $SW \downarrow \prime$ (which corresponds approximately to PAR) and $D_N$, so the role of seasonal PAR fluxes in algal growth remains unclear. More broadly, field studies are required in order to identify precisely how bare ice appearance could impact ice algal assemblages.

If pre-requisites for the initiation of an algal bloom are achieved then an increase in algal biomass is likely, with a concomitant increase in $D_I$. This is consistent with increases in $D_I$ after the first appearance of dark ice, as opposed to 'flickering' between less- and more- dark states. Increases in $D_I$ could be driven by an increase in the spatial extent of ice algal assemblages and/or an increase in algal concentrations per unit area. Previously, Chandler et al. (2015) recorded an increase in 'dirty ice' extent within our common area over an ablation period, though they did not assess algal cell numbers within dirty ice. Whilst algal

concentrations likely increase until a limiting factor becomes apparent, analogous to algal blooms in aquatic systems (Teeling et al., 2016), progressive colonisation of clean ice at the sub-MODIS pixel scale would still result in continued increases in $D_I$ at the regional scale. Indeed, we do not know how much of a given MODIS pixel must be covered in a light algal bloom before the pixel reflectance dips below the dark ice threshold. Given the confounding impacts of cloud cover on MODIS observations, assessing the relative contribution of increases in the extent ($D_E$) versus concentration of algae ($D_I$) on regional variability

in dark ice dynamics is not possible. We suggest, however, that intra-annual patterns in $D_I$ over ablation periods are more consistent with the progression of ice algal blooms than with dynamics in other darkening agents previously discussed.

We interpret our observations as support for the role of ice algae in controlling inter-annual dynamics in darkening, but note that there is currently not sufficient evidence to formally test this assertion. In particular, one major aspect of dark ice variability which ice algae cannot explain is the well-defined maximal spatial extent of dark ice, both in the south-west and

GrIS-wide. Dark ice extent is concentrated spatially into several contiguous areas around the GrIS (Shimada et al., 2016). However, if algal growth were the only factor causing inter-annual variability in dark ice presence, we would expect to see dark ice present wherever the climatological pre-requisites for algal growth are met. These climatological pre-requisites can be found elsewhere, most notably in the $20 \, \mathrm{km}$-wide zone from the ice-sheet margin to the start of the common area examined in this study. This suggests that algal growth controlled by climatology alone cannot fully explain dark ice dynamics in the

south-west sector of the GrIS. In light of our findings, we hypothesise that inter-annual variability in dark ice presence — both in the south-west sector and GrIS-wide — requires (1) melt-out of particulates and (2) blooming of ice algal assemblages. Specifically, we suggest that the in-situ melt-out of particulates defines the spatial extent of dark ice. Algal blooms control dark ice intensity by enhancing the abiotic darkening signal, but only when the climatological pre-requisites for growth are met. For our hypothesis to be correct, melt-out particulates must enable ice algae growth of sufficient magnitude to cause appreciable

darkening, for instance as a source of nutrients.

## 5  Conclusions

We detected hitherto overlooked dynamics of the dark ice zone of south-west of the GrIS using remotely-sensed imagery. Our results show that GrIS dark ice dynamics must be examined across across the full duration of the melt season in order to





understand the processes most likely to be reducing the albedo of bare ice surfaces. We found that in years when the south-west sector of the GrIS darkens, this usually occurs within several days and then remains widespread for the rest of the melt season, indicating that the darkening occurs in response to a common synoptic forcing. The seasons of longest dark ice duration ($D_D$) tend to be associated with earlier retreat of the winter snowpack. Once the ice goes dark then the dark ice intensity ($D_I$) tends

to increase gradually through the melt season. Daily variations in $D_I$ are fairly small.

In our analysis, the JJA sensible heat flux anomaly and the date of bare ice appearance represent the most important climatic controls on dark ice extent ($D_E$), $D_D$ and $D_I$, with higher sensible heat fluxes and earlier bare ice appearance favouring more dark ice. Higher JJA air temperatures and a greater number of days during JJA on which continuous surface melting occurs are also associated with darker years. There is a positive correlation between $D_N$ and the JJA Greenland Blocking Index ($R^2$ 0.46,

$p < 0.05$), which indicates that the climatic conditions which drive darker years can be at least partly attributed to the summer presence of high-pressure blocking systems over the ice sheet.

Our observations suggest that neither deposition of particulates nor cryoconite hole processes can independently explain inter-annual variability in dark ice presence. Our observations tentatively support the proposal that algal blooming is the primary cause of albedo reductions in dark years, likely driven by earlier winter snowpack retreat and positive sensible heat

flux anomalies. However, climatological controls on biological growth alone cannot explain the spatial distribution of inter-annual dark ice presence. We therefore suggest that inter-annual variability in dark ice in the south-west sector of the GrIS is enabled first by the melt-out of particulates. These particulates play an as-yet unknown role in facilitating the growth of ice algal assemblages, which is also controlled by physical/climatic pre-requisites that remain to be identified conclusively.

Future research has several key challenges. First, the spatial distribution, minerology and ice-darkening potential of all melt-

out light-absorbing impurities needs to be quantified. Second, the spatial distribution and hence ice-darkening potential of ice algae needs to be examined not just at plot scales but also at scales of hundreds of metres and more. Third, if algal cells are found to be abundant and to be the primary driver of dark ice, then the physical/climatic and nutrient controls on the growth of ice algae need to be established. Last, all these findings should be assimilated into a physical model of ice surface albedo that can be embedded within a regional climate model, in order to project the impact of dark ice upon runoff from the GrIS during

the 21$^{\text{st}}$ century.

## 6  Code availability

The Modèle Atmosphérique Régional (MAR) is an open-source regional climate model. Source code of MARv3.6.2 is available at ftp://ftp.climato.be/fettweis/MARv3.6/.src/.





## 7 Data availability

Monthly outputs from MAR are available at ftp://ftp.climato.be/fettweis/MARv3.6.2/ for different model domains and resolutions. If daily outputs are required, please email xavier.fettweis@ulg.ac.be. MODIS data are available from the USGS LPDAAC Data Pool (https://lpdaac.usgs.gov/data_access/data_pool).

## Appendix A: Choice of spectral thresholds

We validated the spectral thresholds used in this study through comparison to hemispherical-conical reflectance factor (HCRF) measurements made in the field on 19 July 2016 in the vicinity of S6 (67º4′28.6″ N, 49º21′32.4″ W). We made HCRF measurements for three qualitatively identified surface types: (1) white ice, (2) light algal bloom (characterised by a light brown colouration to the ice surface) and (3) heavy algal bloom (characterised by a dark brown colouration to the ice surface). The measurements were made following the HCRF measurement protocol described by Cook et al. (in review, 2017). Briefly, an ASD Field Spec Pro spectral radiometer with an 8 degree fore-optic was positioned 30 cm above the sample surface with a nadir viewing angle. This device measures reflected radiance in the wavelength range 350–2500 nm and therefore senses reflected radiance over about 95% of the solar spectrum. The sample surface was qualitatively homogenous in a buffer zone of at least 30 cm around the viewing footprint of the sensor. We calculated the mean of at least twenty sample replicates, all of which were made within one minute without changing the sensor position. All measurements were acquired within a 2 h sampling window around solar noon, thereby minimising error due to changing solar zenith. The sky was cloud-free throughout the measurement window. Naturally-illuminated nadir-view HCRF is reported for consistency with the reported MOD09GA data.

The field spectra (Fig. A1) show that the bare-ice threshold used by Shimada et al. (2016) adequately captures white-ice surfaces. Their threshold of $R_{620-670nm} < 0.4$ to define dark ice is conservative and prevents positive identification of light algal blooms. In this study we used a threshold of $R_{620-670nm} < 0.45$, which is set to just below our field observations of light algal bloom reflectance in order to reduce the likelihood of false positives.

*Author contributions.* A.T., J.B. and M.T. designed the study. A.T. processed the MODIS data, carried out most of the analysis and interpretation and wrote most of the manuscript. J.C. and C.W. contributed to the interpretation and wrote parts of the manuscript. X.F. developed MAR and provided the model outputs. A.H. undertook additional energy balance analysis and contributed to the interpretation. All authors discussed the findings and commented on the manuscript.

*Competing interests.* The authors declare no competing financial interests. J. Bamber is Advisory Editor of The Cryosphere.

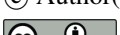



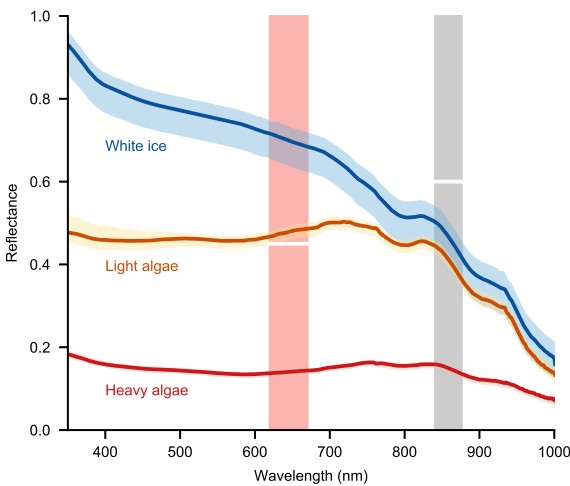

**Figure A1.** Field HCRF spectra acquired on the GrIS (see Appendix A). For each surface type, solid lines denote mean reflectance and the shaded bounds are delimited by the minimum and maximum reflectances. The gray shaded box corresponds to MODIS Band 2 (841–876 nm), and the red shaded box to MODIS Band 2 (620–670 nm). White divisions in each box correspond to the spectral thresholds utilised in this study to define bare and dark ice areas.

*Acknowledgements.* This study was supported by the UK Natural Environment Research Council Consortium Grant 'Black and Bloom' (NE/M021025). In addition to the authors, the Black and Bloom team comprises A. Anesio, L. Benning, E. Hanna, S. Hofer, A. Holland, T. Irvine-Fynn, S. Lutz, J. McCutcheon, J. McQuaid, M. Nicholes, E. Sypianska, C. Williamson and M. Yallop.





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
