# Peer review of "Dark ice dynamics of the south-west Greenland Ice Sheet"

_The Cryosphere, 2017_

## Referee Comment (RC1) · Anonymous Referee #1 · 27 Jun 2017

This is an informative study that combines remote sensing measurements of albedo with regional climate modeling to identify some of the factors that are associated with dynamics of the dark ice zone in southwest Greenland. The study does not offer any definitive conclusions about the actual processes governing these dynamics. But given that our understanding of biological controls on surface ice albedo is in its infancy, I think the associations between variables that are described here constitute a worthwhile contribution to the literature. The paper is quite well-written and includes insightful, if sometimes rather speculative, discussion.

The issues I describe below may require a bit of attention, though they are generally minor. I should add that a very similar remote sensing analysis was presented by Shimada et al (2016), and it seems important that the authors of that study should

review and comment on this study. From my perspective, the present study seems to adequately describe its results within the context of Shimada et al. Furthermore, a novel component of the present study is that it combines regional climate simulations with the remote sensing analysis.

General issues:

The fact that the JJA melt-out-flux (MOF) is universally negative (Figure 4C) leads me to question the utility of this quantity. It is argued that when this quantity is positive conditions are favorable for melt-out of particles and unfavorable for cryoconite hole formation. But since the quantity is always negative during the summer, and since there is evidence (?) for melt-out of particles during summer, this quantity does not appear to be a good predictor of melt-out conditions. If this reasoning seems sound, I suggest that the authors consider removing this quantity altogether from the manuscript.

Sensible heat flux is deemed to be an important correlated variable with dark ice dynamics. How closely does the sensible heat flux track near-surface (or lower tropospheric) air temperature? They may be closely linked over Greenland. Sensible heat flux should loosely track (1) the difference in temperature between the air and surface, and (2) the near-surface wind speed. Since the ice surface is always at 0C when melting, the temperature difference is governed exclusively by air temperature. It is unclear, though, how important the wind speed is.

The first paragraph of Data and Methods indicates that both MOD09GA and MOD10A1 MODIS reflectance/albedo products are used, but it is not clear to me which analyses and sections of the paper use which products. Please clarify this. Is MOD10A1 perhaps a derived product from MOD09GA, and it is really the former that is applied here? If so, please clarify this. Secondly, why is MOD10A1 used instead of other MODIS albedo product(s), like MCD43 for instance? Thirdly, please describe the native resolution of the MODIS data applied in this study.

p.4, line 29: "It is also noteworthy that R620-670nm straddles a transition zone be-

tween wavelengths mostly influenced by LAIs and wavelengths mostly influenced by grain evolution and interstitial water." - In that case, why is this wavelength chosen to discriminate dark ice (as darkened by LAIs), instead of a shorter wavelength?

The definition of intensity ($D_I$) given on p.5 is slightly unclear to me. Is $D_I$ the average reflectance over the entire common area, or the average reflectance of the "dark" pixels within the common area? If it is the former, then $D_I$ is affected both by the extent and the darkness of the dark ice, and it is therefore not independent of $D_E$. Please clarify this.

The term "melt-out", as in "melt-out of particulates" is used frequently in this manuscript, but the precise meaning or process indicated by this term was at times unclear to me. I suggest clearly describing what is meant by "melt-out", at least at the first instance of its use.

Minor comments:

p2, line 6: "Surface melting is controlled primarily by albedo" - I agree, but it would bolster your case to include one or more references in support of this claim.

p2, line 18: "The GrIS-wide bare-ice ablation zone extent increased by 4.4% per year..." - Is this a relative or absolute (as in percent of whole ice sheet) change? I assume the former, but please clarify.

p3, line 22: Is it necessary that the cryoconite reside beneath a layer of meltwater for the albedo increase to occur? Perhaps the melt layer augments the change, but I suspect the hole depth is the more important factor for hemispheric albedo increase. You might want to add nuance to this statement.

p4, line 28: "precisely identify precisely"

p.5, line 18: "Only days in which at least 50% of the common area was cloud-free were included in the calculation" - And furthermore, were only cloud-free pixels used in this average? I assume so, but please clarify.
p.6, line 4: "... equal-area 7.5 x 7.5 km..." - Earlier it is stated that model pixels are 600m x 600m. Please rectify this.

Equation 4: It appears that SHF is defined as positive into the surface, but please confirm.

p.7, line 13: "were been"

Figure 2 caption: "... the entire common area had D_I < 0.45." - Just to be sure, do you mean that every pixel in the common area had D_I < 0.45 (as communicated) or that the average D_I of the common area was less than 0.45?

Figure 2: What do the black triangles represent?

p.12, line 10: " ...The only published measurements of black carbon on the GrIS are from the north-west (Aoki et al, 2014; Polashenski et al, 2015)" - This statement needs refining, as there have been BC measurements from elsewhere on Greenland, including, e.g., by McConnell et al (2007, doi:10.1126/science.1144856) and Doherty et al (2013, doi:10.1002/jgrd.50235).

p.13, line 2: missing citation

p.14, line 23-25: Please see general comment about relationship between air temperature and SHF. I am wondering if the two quantities referenced in this sentence are closely related to each other. If so, it would be worth commenting on that here.

p.15, line 14: "... versus concentration of algae (D_I)..." - Related to my earlier comment, is D_I a true measure of algae concentration, or is it also affected by the extent of the dark zone?

p.15, line 35: "across across"

In the figure captions, please describe the variables in addition to using their symbols.

---

## Referee Comment (RC2) · Anonymous Referee #2 · 21 Jul 2017

See attached supplement.

Please also note the supplement to this comment:
https://www.the-cryosphere-discuss.net/tc-2017-79/tc-2017-79-RC2-supplement.pdf

[Figure]

**Dark ice dynamics of the south-west Greenland ice sheet**

A. J. Tedstone, J. L. Bamber, J. M. Cook, C. J. Williamson, X. Fettweis, A. J. Hodson, and M. Tranter

**Summary**

MODIS satellite imagery is used to examine fluctuations in the extent of impurity-rich bare ice (dark ice) along the western margin of the Greenland Ice Sheet. A threshold on MODIS blue and red reflectance is used to identify bare ice and dark ice. Potential drivers of bare ice variability are examined using outputs of the MAR regional climate model, including shortwave radiation, longwave radiation, and sensible heat flux, in an attempt to understand causes of variability. The authors argue that while outcropping particulates are a major factor in bare ice albedo variability, the presence of biological organisms may also play an important role.

**General Comments**

The topic covered by the paper is important to our understanding of factors contributing to fluctuations in the albedo of impurity-covered ice in the ablation area of the Greenland ice sheet. It overlaps somewhat with the recent study of Shimada et al. (2016), but extends the analysis to a full summer season and attempts to understand drivers of dark ice variability.

I feel the authors need better support for their arguments that biology is a major driver of bare ice albedo variability. There is no definitive proof for this and I don't think the authors have successfully ruled out melt-out of impurities, sub-grid scale variability in snow cover and/or superimposed ice, or even the presence of liquid water, as potential causes of the variability. The authors have suggested that microorganisms appear to require the presence of outcropping material at the surface. If this is the case on a large scale, outcropping dust should control local and inter-annual variations in albedo as well. The authors' arguments that local-scale variability in dark ice extent can be explained not by dust melt-out, but by microorganisms, is inconsistent with the apparent need for dust as a microbial nutrient source on a larger scale.

I think that much of the variability the authors attribute to microorganisms could be attributed to dynamics of melt-out at small scales instead. Inter-annual variations in dark ice extent can be explained by the presence of superimposed ice, perhaps not fully accounted for in MAR. Increases in "Dark Ice Intensity" over time could be related to changes in surface cover within a relatively large MODIS grid box as snow patches and areas of superimposed ice melt away, exposing dark material beneath. The fact that sensible heat flux is a relatively important factor, as is the number of days where temperature is greater than zero suggests that melting of snow and ice could be an important factor independent of biological organisms.

Therefore, there appears to be insufficient information to state definitively the cause of the variations in dark ice extent and intensity, although I think the authors have shown that local deposition from year to year can probably be ruled out as a contributing factor.

**Fig. 1.**

**Supplement:**

**Dark ice dynamics of the south-west Greenland ice sheet**

A. J. Tedstone, J. L. Bamber, J. M. Cook, C. J. Williamson, X. Fettweis, A. J. Hodson, and M. Tranter

**Summary**
MODIS satellite imagery is used to examine fluctuations in the extent of impurity-rich bare ice (dark ice) along the western margin of the Greenland Ice Sheet.  A threshold on MODIS blue and red reflectance is used to identify bare ice and dark ice.  Potential drivers of bare ice variability are examined using outputs of the MAR regional climate model, including shortwave radiation, longwave radiation, and sensible heat flux, in an attempt to understand causes of variability.  The authors argue that while outcropping particulates are a major factor in bare ice albedo variability, the presence of biological organisms may also play an important role.

**General Comments**
The topic covered by the paper is important to our understanding of factors contributing to fluctuations in the albedo of impurity-covered ice in the ablation area of the Greenland ice sheet.  It overlaps somewhat with the recent study of Shimada et al. (2016), but extends the analysis to a full summer season and attempts to understand drivers of dark ice variability.

I feel the authors need better support for their arguments that biology is a major driver of bare ice albedo variability.  There is no definitive proof for this and I don't think the authors have successfully ruled out melt-out of impurities, sub-grid scale variability in snow cover and/or superimposed ice, or even the presence of liquid water, as potential causes of the variability.  The authors have suggested that microorganisms appear to require the presence of outcropping material at the surface.  If this is the case on a large scale, outcropping dust should control local and inter-annual variations in albedo as well.  The authors' arguments that local-scale variability in dark ice extent can be explained not by dust melt-out, but by microorganisms, is inconsistent with the apparent need for dust as a microbial nutrient source on a larger scale.

I think that much of the variability the authors attribute to microorganisms could be attributed to dynamics of melt-out at small scales instead.  Inter-annual variations in dark ice extent can be explained by the presence of superimposed ice, perhaps not fully accounted for in MAR.   Increases in "Dark Ice Intensity" over time could be related to changes in surface cover within a relatively large MODIS grid box as snow patches and areas of superimposed ice melt away, exposing dark material beneath.  The fact that sensible heat flux is a relatively important factor, as is the number of days where temperature is greater than zero suggests that melting of snow and ice could be an important factor independent of biological organisms.

Therefore, there appears to be insufficient information to state definitively the cause of the variations in dark ice extent and intensity, although I think the authors have shown that local deposition from year to year can probably be ruled out as a contributing factor.

Given a lack of clear evidence supporting a biological source for inter-annual and intra-annual variability in bare ice albedo, I feel that the authors should reduce the emphasis on biological organisms as a source of variability and should also give credence to the possibilities mentioned above.

The authors should also address the possibility that the thresholds used here can falsely identify liquid water and possibility even snow or firn as ice or dark ice. The first is probably a minor factor, but the second could potentially lead to a misinterpretation of the results.

The work presented here provides a valuable investigation of variations in ice albedo and the presence of impurities in the ablation area of the Greenland ice sheet. I support publication of the study, provided the authors address the points provided in this review.

**Specific Comments**
**P. 1, Line 1:** The recent increases in runoff are not caused by reduced albedo but by changes in atmospheric circulation and atmospheric warming. Albedo changes resulting from these changes amplify melt. Please clarify.
**P. 1, Line 7:** Add "in the future" after "will evolve".
**P. 2, Line 6:** The statement that "surface melting is controlled by albedo" should be clarified. Other components of the energy balance certainly play a role in controlling melting. Albedo can only play a role with sufficient downward shortwave radiation. Melting can potentially occur during portions of the year when there is less solar radiation as a result of sensible, or longwave fluxes. Please revise this statement, e.g. "Surface albedo plays an important role in modulating surface melt as the surface darkens with warming temperatures…."
**P. 3, Line 34 – P. 4, Line 4:** Is there a reference to which the authors can refer here or are these unpublished results of the authors? Please clarify the source in the text.
**P. 4, Line 8:** Independent of these processes, there is also the possibility of consolidation of impurities at the surface due to melt, which the authors do discuss later in the manuscript. Perhaps change "inorganic particulate deposition" to "inorganic particulate deposition or redistribution".
**P. 4, Lines 28-29:** These are all good points, but perhaps now say what the authors think *can* be done using the thresholds used here.
**P. 4, Lines 29-30:** Are the authors saying that some of the variability in extent or intensity could then be associated with grain size evolution and the presence of water? Please clarify.
**P. 5, Line 1:** Clarify how the maximum area was defined, e.g. using daily MODIS reflectance values.
**P. 5, Line 8:** Explain why pixels 1 km from the ice sheet margin were removed.
**P. 5, Lines 11-12:** What is meant by "all the pixels", the number of pixels or fraction of pixels?
**P. 5, Line 13:** Clarify that this is the percentage of all daily cloud-free observations that were classified as "dark" in each JJA period.

**P. 5, Line 15-16:** It is a bit confusing to refer to this as intensity and to have a lower number indicate a larger intensity. Can't this just be referred to as the average reflectance? Then a lower reflectance is associated with a darker surface.

**P. 6, Line 6:** Include a reference for the ECMWF reanalysis: (Dee et al., 2011) doi:10.1002/qj.828

**P. 6, Line 15:** Is the daily energy for melt-out "MOF"? Define MOF here. Based on the authors statements it doesn't seem that the MOF is necessarily a proven measure of the conditions needed to produce melt-out. If so it should be made clear that the MOF is suggestive of the conditions needed to cause melt-out, but does not necessarily indicate whether melt-out is occurring or not.

**P. 7, Line 4:** Clarify that this "extension is relative to the study of Shimada et al. (2016), which only examined July.

**P. 7, Line 8:** Change "time lag…" to "time lag between $t_B$ and the first identified occurrence of dark ice of 10-15 days".

**P. 7, Line 10:** Anticyclonic days don't seem to be shaded gray in Fig. 4.

**P. 9, Line 6:** Change "magnitude of dark ice" to something like "extent and intensity of dark ice" or "extent and reflectivity of dark ice".

**P. 9, Line 8:** Clarify "years when the ice went dark". Perhaps "years when $D_E$ was higher" would be more specific.

**P. 9, Line 22:** Change "Not only was winter snowfall" to "Not only was 2014-2015 winter snowfall…" for clarity.

**P. 11, Line 24:** Briefly not how the weathering crust forms.

**P. 12, Line 27:** Should "decimeter" be "decameter"?

**P. 12, Line 21 – P. 13 Line 2:** I am not totally convinced by this argument. Much of this could be explained by the presence of superimposed ice, sub-grid scale exposure of bare ice, or even the presence of firn that is mis-classified as bare ice. I don't think the authors can rule out melting as a primary cause of the observed variability, especially since they do not utilize measurements or estimates of melt here. I think the authors should be more careful to acknowledge that melt could be responsible for the observed variability, but that the results also suggest that other factors could be involved.

**P. 13, Lines 17-25:** The variability the authors are discussing seems consistent with the hypothesis of Shimada et al. (2016) except with regard to the changes in dark ice intensity during 2012 and between 2012 and 2013. The statement that "our results reveal a different spatio-temporal pattern" is therefore a bit confusing. As for previous section, the changes in intensity during 2012 could be explained by sub MODIS-grid-scale processes such as melting of snow patches, collecting meltwater. 2012 was a high melt year while 2013 was a low melt year. During 2013, ice is exposed for a much shorter length of time, and the presence of superimposed ice, or again, patches of snow covering the ice could explain the lack of dark ice during that year.

**P. 17, Lines 8-9:** The surface must be a mixture of impurities and biological materials, or could even be abiotic. How is the material assumed to be algae?

**Figure 1:** It would be useful for the reader to include numbers indicating the value of $D_E$ for each image.

**Figure 2:** Mention $t_B$ in the caption.

**Figure 3:** Note that the snow depth is from MAR. It would be interesting to also see $t_B$ in this figure, to allow for a comparison with MAR.

**Technical Corrections**
**P. 4, Line 15:** Change "cloud" to "clouds".
**P. 4, Line 28:** Change "precisely identify precisely" to "precisely identify"
**P. 5, Line 18:** Add "($D_N$)" after "normalized darkness" for clarity.
**P. 5, Line 27:** The phrase "with any…only allowed to be cloudy" is confusing. Perhaps just change to "excluding cloudy days".
**P. 6, Line 12:** Place a parenthesis around ($T>0$) for clarity.
**P. 6, Line 13:** Change "were been" to "were"
**P. 9, Line 3:** Change "not explicable by" to "cannot be explained by"
**P. 9, Line 14:** Change "snowfall which occurs" to "snowfall that occurs"

---

## Author Comment (AC1) · 17 Aug 2017

**Response to reviews**

**'Dark-ice dynamics of the south-west Greenland ice sheet' by A. J. Tedstone *et al.***

Dear Prof. Tedesco,

We would like to thank both referees for taking the time to make detailed comments, which have resulted in a much-improved manuscript. We have taken care to add nuance to several sections of the manuscript. We respond inline to the referee comments below. Referee comments are in *italic* and changes in the manuscript are in **bold**.

We hope that you find our revisions make our manuscript suitable for publication in *The Cryosphere.*

Yours sincerely,

Andrew Tedstone, on behalf of the co-authors.

RC1

*This is an informative study that combines remote sensing measurements of albedo with regional climate modeling to identify some of the factors that are associated with dynamics of the dark ice zone in southwest Greenland. The study does not offer any definitive conclusions about the actual processes governing these dynamics. But given that our understanding of biological controls on surface ice albedo is in its infancy, I think the associations between variables that are described here constitute a worthwhile contribution to the literature. The paper is quite well-written and includes insightful, if sometimes rather speculative, discussion.*

*The issues I describe below may require a bit of attention, though they are generally minor. I should add that a very similar remote sensing analysis was presented by Shimada et al (2016), and it seems important that the authors of that study should review and comment on this study. From my perspective, the present study seems to adequately describe its results within the context of Shimada et al. Furthermore, a novel component of the present study is that it combines regional climate simulations with the remote sensing analysis.*

*General issues:*

*The fact that the JJA melt-out-flux (MOF) is universally negative (Figure 4C) leads me to question the utility of this quantity. It is argued that when this quantity is positive conditions are favorable for melt-out of particles and unfavorable for cryoconite hole formation. But since the quantity is always negative during the summer, and since there is evidence (?) for melt-out of particles during summer, this quantity does not appear to be a good predictor of melt-out conditions. If this reasoning seems sound, I suggest that the authors consider removing this quantity altogether from the manuscript.*

The cryoconite hole melt-out process was hypothesised as a driver of inter-annual variability in dark ice extent by Shimada et al (2016). The only evidence in the literature related to this hypothesis in the form of field measurements made in the south-west GrIS ablation zone by Chandler et al (2015)

**(which we cite more extensively in the revised manuscript)**. These field measurements covered only a single season, 2015. Briefly, they presented limited evidence of cryoconite hole melt-out during a few days of warm, cloudy conditions, in which a few holes melted out but spatial coverage of cryoconite holes remained high. This event occurred in the context of an overall trend of increasing cryoconite hole coverage through the season. Hence, evidence for melt-out of cryoconite holes in summer is equivocal as the only field measurements from the GrIS to date show that melt-out does occur but not necessarily with widespread spatial impact.

We therefore tried to characterise the likelihood of cryoconite hole melt-out over wider spatial scales by defining the MOF quantity, which attempts to characterise 'warm, cloudy conditions' by considering the importance of sensible and longwave heat fluxes against shortwave fluxes. We note that we cannot test this experimental quantity (derived from a regional climate model) against field measurements as Chandler et al did not measure the full energy balance. We further assessed the potential for precipitation events to cause cryoconite hole melt-out, in case these events were not captured by MOF.

In summary, as existing evidence for the melt-out of cryoconite holes and associated climate conditions is equivocal, the MOF analysis is a key element of our analysis.

*Sensible heat flux is deemed to be an important correlated variable with dark ice dynamics. How closely does the sensible heat flux track near-surface (or lower tropospheric) air temperature? They may be closely linked over Greenland. Sensible heat flux should loosely track (1) the difference in temperature between the air and surface, and (2) the near-surface wind speed. Since the ice surface is always at 0C when melting, the temperature difference is governed exclusively by air temperature. It is unclear, though, how important the wind speed is.*

To examine this suggestion in further detail we use daily timeseries (average for the common area) of wind speed [UV], air temperature [TT], minimum air temperature [TTMIN] and sensible heat flux [SHF].

There is a strong correlation between TT and SHF ($R^2$ 0.54, $p < 0.01$), and between TT and UV ($R^2$ 0.67, $p < 0.01$). A multiple regression model of TT+UV~SHF also shows high correlation ($R^2$ 0.80, $p < 0.01$). Imperfect correlations are to be expected given the averaging over the common area. This analysis indicates that both TT and UV are important in driving high SHF into the ice sheet surface.

[Figure]

Response Figure 1. JJA minimum daily near-surface air temperature versus sensible heat flux, averaged over the common area. Both variables output by MAR.

However, we also note that, during JJA, positive daily TTMIN only occurs on days when SHF modelled at 12:00 is positive (Response Figure 1). We do not know the minimum daily SHF so cannot test for an association here, but nevertheless these results suggest a relationship between positive SHF and above-zero TTMIN.

Comparison of daily TTMIN and wind speed (Response Figure 2) suggests that, in general, positive TTMIN only occurs routinely at wind speeds in excess of ~6 m s$^{-1}$, which is also represented as a histogram in Response Figure 3. Thus, this suggests that higher wind speeds are the principal cause of higher SHF (i.e. into the ice sheet surface), and this higher SHF in turn makes positive TTMIN more likely.

[Figure]

Response Figure 2. For JJA, daily minimum near-surface air temperature versus daily wind speed in the common area. Both variables output by MAR.

[Figure]

Response Figure 3. Histograms of daily wind speed in common area when the daily minimum temperature is below 0°C (green) compared to above 0°C (blue).

We have summarised these findings in the revised manuscript in **Sect. 3.2.2:**

> **Over daily timescales, higher SHF was associated with warmer near-surface air temperatures ($R^2$ 0.54, p < 0.01) but more strongly with higher near-surface wind speeds ($R^2$ 0.67, p < 0.01). Days on which the minimum air temperature was greater than 0 °C had mean wind speeds of 6.5 +-1.8 m s$^{-1}$ +- 1 sigma), compared to 4.9 +- 1.3 m s$^{-1}$ +- 1 sigma} on days when the minimum air temperature was 0 °C or less.**

And have noted in the Conclusions that **higher SHF was associated with higher wind speeds.**

*The first paragraph of Data and Methods indicates that both MOD09GA and MOD10A1 MODIS reflectance/albedo products are used, but it is not clear to me which analyses and sections of the paper use which products. Please clarify this. Is MOD10A1 perhaps a derived product from MOD09GA, and it is really the former that is applied here? If so, please clarify this. Secondly, why is MOD10A1 used instead of other MODIS albedo product(s), like MCD43 for instance? Thirdly, please describe the native resolution of the MODIS data applied in this study.*

MOD10A1 is a standalone product which is produced separately to MOD09GA. MOD10A1 is a daily albedo product, unlike MCD43 which is a multi-day composite product. In terms of our dark ice dynamics observations, the only part of the MOD10A1 product that we use is the cloud discrimination layer in order to mask our cloudy pixels in MOD09GA. However, we do use MOD10A1 albedo to compute SW$_{net}$ as part of our MOF analysis (Sect. 2.5).  The nominal resolution of MODIS sinusoidal gridded products is 500 m which we now note as follows:

> Both MODIS Level-2 products are delivered on a sinusoidal grid **at 500 m nominal resolution** which causes significant distortion…

*p.4, line 29: "It is also noteworthy that R620-670nm straddles a transition zone between wavelengths mostly influenced by LAIs and wavelengths mostly influenced by grain evolution and interstitial water." - In that case, why is this wavelength chosen to discriminate dark ice (as darkened by LAIs), instead of a shorter wavelength?*

620-670 nm is within the visible range and therefore predominantly affected by LAIs rather than grain evolution. Effects from grain evolution are likely minor at an upper bound of 670 nm; the statement noted above was partly due to an internal miscommunication, where the upper bound was accidentally thought to be ~700 nm.

Individual contaminants may alter the reflectance at specific wavelengths within the blue and green parts of the spectrum, compared to decreasing influence at longer visible wavelengths. For instance, heavy loading of certain dusts on snow can reduce the reflectance in the blue wavelengths but leave the green-red part of the spectrum reflecting efficiently (e.g. Skiles et al, 2017, *J. Glaciology*). As such, the red part of the spectrum is a better indicator of dark ice than the MODIS blue or green bands, which could lead to erroneous capture of dark snow as dark ice if the dust loading is high enough. We also note that our empirical evidence, in the form of field spectra presented in Appendix A, adequately show that thresholding 620-670 nm captures light and heavy algal blooms.

We have modified Sect. 2.1 to better reflect these factors:

> **However, we note that we do not know precisely what this dark ice threshold represents physically. The red band (620-670 nm) sits within the visible wavelengths and is therefore affected mostly by LAIs rather than grain evolution or water ponding, which mostly affect the near-infrared wavelengths (700-1100 nm). However, we caveat that other mechanisms can also reduce the reflectance across the entire solar spectrum, including in the red waveband. These include reduction in volume-scattering due to wind or water 'polishing' the ice surface, infilling of interstitial air spaces with meltwater, and 'trapping' by roughness features such as crevasses. Nevertheless, by combining these thresholds we are able to distinguish at first order between clean and dark (LAI-laden) ice surfaces.**

*The definition of intensity (D_I) given on p.5 is slightly unclear to me. Is D_I the average reflectance over the entire common area, or the average reflectance of the "dark" pixels within the common area? If it is the former, then D_I is affected both by the extent and the darkness of the dark ice, and it is therefore not independent of D_E. Please clarify this.*

$D_I$ is the average reflectance over the entire common area. We have clarified as follows:

> Third, intensity ($D_I$) was defined as the mean daily reflectance over 620-670 nm **of all cloud-free pixels in the common area**, and annual intensity as the mean of all cloud-free days in each JJA period.

*The term "melt-out", as in "melt-out of particulates" is used frequently in this manuscript, but the precise meaning or process indicated by this term was at times unclear to me. I suggest clearly describing what is meant by "melt-out", at least at the first instance of its use.*

In response to this comment we have **modified the manuscript so that the term 'melt-out' is used only in reference to the melt-out of cryoconite holes**. We no longer use this term when discussing ablating ancient ice as a source of particulates.

*Minor comments:*

*p2, line 6: "Surface melting is controlled primarily by albedo" - I agree, but it would bolster your case to include one or more references in support of this claim.*

Please see response to RC2.

*p2, line 18: "The GrIS-wide bare-ice ablation zone extent increased by 4.4% per year..." - Is this a relative or absolute (as in percent of whole ice sheet) change? I assume the former, but please clarify.*

On closely re-reading Shimada et al (2016) we find that it is neither option. Instead, the 4.4% per year is of the mean bare ice extent over 2000-2014. We have therefore rephrased as follows:

> The GrIS-wide bare-ice ablation zone extent increased by **7,158 km$^2$** per year on average from 2000 to 2014, although with substantial inter-annual variability of between 5 % **(89,975 km$^2$)** and 16 % **(279,075 km$^2$)** of the ice sheet **surface (Shimada et al., 2016)**.

*p3, line 22: Is it necessary that the cryoconite reside beneath a layer of meltwater for the albedo increase to occur? Perhaps the melt layer augments the change, but I suspect the hole depth is the more important factor for hemispheric albedo increase. You might want to add nuance to this statement.*

No it is not necessary, the reviewer is correct to suggest that the majority of the albedo increase is due to hiding the cryoconite at depth in the weathering crust; however, specular reflection from the water surface will enhance the effect by preventing incoming light from being 'trapped' by multiple reflection within the hole. We have edited the text as follows:

> Hole formation increases the albedo relative to dispersed cryoconite by sequestering the low-albedo cryoconite from the ice surface at depth, **resulting in a hemispheric albedo increase that will be further enhanced by specular reflection** when covered by a reflective layer of meltwater (Boggild et al, 2010).

*p4, line 28: "precisely identify precisely"*

Thank you, **corrected**.

*p.5, line 18: "Only days in which at least 50% of the common area was cloud-free were included in the calculation" - And furthermore, were only cloud-free pixels used in this average? I assume so, but please clarify.*

Correct, we have clarified as follows:

> Only days in which at least 50 % of the common area was cloud-free were included in the calculation, **and only the cloud-free pixels within the common area were used.**

*p.6, line 4: "... equal-area 7.5 x 7.5 km..." - Earlier it is stated that model pixels are 600m x 600m. Please rectify this.*

Our phrasing was confusing here. MODIS data are at 600 m, MAR at 7.5 km, but when we compare MODIS data with MAR outputs we bin MODIS data into MAR pixels:

> …yielding a spatial resolution of ~ 600 x 600 m. **When undertaking comparisons with MAR outputs, cloud-free MODIS data were binned into 7.5 km pixels to match MAR's resolution.**

*Equation 4: It appears that SHF is defined as positive into the surface, but please confirm.*

Yes, this is correct and **we now note this in Sect. 2.5** (Meteorological and climatological data).

*p.7, line 13: "were been"*

Thanks, **corrected.**

*Figure 2 caption: "... the entire common area had D_I < 0.45." - Just to be sure, do you mean that every pixel in the common area had D_I < 0.45 (as communicated) or that the average D_I of the common area was less than 0.45?*

Correct, caption updated:

> Black squares denote days on which the **average D_I of the cloud-free common area was < 0.45.**

*Figure 2: What do the black triangles represent?*

The black triangle represents the date of snow clearing $\tilde{t_B}$, as defined in the legend located in the upper-right of Figure 2. For clarity we have also added an explanation to the caption:

> **Black triangles denote the date of snow clearing, \tilde{t_B}.**

*p.12, line 10: " ...The only published measurements of black carbon on the GrIS are from the north-west (Aoki et al, 2014; Polashenski et al, 2015)" - This statement needs refining, as there have been BC measurements from elsewhere on Greenland, including, e.g., by McConnell et al (2007, doi:10.1126/science.1144856) and Doherty et al (2013, doi:10.1002/jgrd.50235).*

Thank you for drawing our attention to additional BC measurements in the literature. We have refined our statement to focus on surface (as opposed to ice-core) samples:

Measurements of black carbon **in snow on the present-day surface** of the GrIS **(as opposed to in ice cores)** have been made in the north-west (Aoki et al 2014, Polashenski et al 2015) **or high in the accumulation zone (Hegg et al 2010, Doherty et al 2013)**. However, at only a few ppb, these measurements of black carbon are insufficient to explain the substantial reduction in reflectance in the south-west (Shimada et al 2016).

*p.13, line 2: missing citation*

Apologies, a typo crept in at the last moment before submission, the citation is **Chandler et al. (2015, TC).**

*p.14, line 23-25: Please see general comment about relationship between air temperature and SHF. I am wondering if the two quantities referenced in this sentence are closely related to each other. If so, it would be worth commenting on that here.*

Please see our response to the associated general comment, above.

*p.15, line 14: "... versus concentration of algae (D_I)..." - Related to my earlier comment, is D_I a true measure of algae concentration, or is it also affected by the extent of the dark zone?*

D_I is purely a measure of how dark the entire common area is. The spatial extent of the common area is fixed for the entire duration of the study and so D_I is therefore independent of the extent of the dark zone. See also our response to earlier comment.

*p.15, line 35: "across across"*

Thank you, **corrected.**

*In the figure captions, please describe the variables in addition to using their symbols.*

**We now describe the variables**, or where the description would be excessively long, provide a reference back to where they are first defined.

RC2

*Summary*

*MODIS satellite imagery is used to examine fluctuations in the extent of impurity-rich bare ice (dark ice) along the western margin of the Greenland Ice Sheet. A threshold on MODIS blue and red reflectance is used to identify bare ice and dark ice. Potential drivers of bare ice variability are examined using outputs of the MAR regional climate model, including shortwave radiation, longwave radiation, and sensible heat flux, in an attempt to understand causes of variability. The authors argue*

*that while outcropping particulates are a major factor in bare ice albedo variability, the presence of biological organisms may also play an important role.*

*General Comments*

*The topic covered by the paper is important to our understanding of factors contributing to fluctuations in the albedo of impurity-covered ice in the ablation area of the Greenland ice sheet. It overlaps somewhat with the recent study of Shimada et al. (2016), but extends the analysis to a full summer season and attempts to understand drivers of dark ice variability.*

*I feel the authors need better support for their arguments that biology is a major driver of bare ice albedo variability. There is no definitive proof for this and I don't think the authors have successfully ruled out melt-out of impurities, sub-grid scale variability in snow cover and/or superimposed ice, or even the presence of liquid water, as potential causes of the variability.*

We are sympathetic to the concerns that referee 2 raises. At no point do we argue that we have definitive proof of '*biology*' constituting '*a major driver of bare ice albedo variability*'; the basis of this paper was rather to identify the most likely driving mechanisms of ice darkening. In conjunction with the existing literature our observations suggest that the most likely driving mechanism is biology, or, more specifically, algal growth. The referee goes on to discuss each of their other listed processes in more depth so please see our responses inline.

*The authors have suggested that microorganisms appear to require the presence of outcropping material at the surface. If this is the case on a large scale, outcropping dust should control local and inter-annual variations in albedo as well. The authors' arguments that local-scale variability in dark ice extent can be explained not by dust melt-out, but by microorganisms, is inconsistent with the apparent need for dust as a microbial nutrient source on a larger scale.*

We disagree. There is likely to be a significant difference between [A] the composition (not necessarily aborptive in visible spectrum) and concentration (relatively low) of outcropping materials required as an input to algal growth, for instance to supply nutrients, versus [B] the composition (absorptive in visible spectrum)  and concentration (high) of outcropping materials necessary to cause a reduction in surface reflectance (e.g. Warren, 2013, *JGR*). The other pre-requisites of algal growth – meltwater presence and PAR – are then controlled by the meteorology of each melt season. Furthermore, we reiterate that we are unable to explain our temporal observations of dark ice dynamics by any known inorganic process (see discussion in Sect. 4.1), but that they do fit with darkening caused by the procession of algal growth.

Of course, over periods of several decades if outcropping dust is a fundamental pre-requisite to algal darkening then we would ultimately expect fluctuations in the dust supply rate to govern dark ice dynamics. However, our observations in this manuscript imply that over inter-annual timescales outcropping dust as a source of nutrients can be relied upon from one year to the next. Further examination of this topic is beyond the reach of our study, which is why we call for future field studies to quantify the distribution, mineralogy and ice-darkening potential of outcropping materials.

**We have made various additions to the text, especially in Sect. 4.3, and also added some additional discussion to the end of Sect. 4.3 which draws on our response above.**

See also our response to RC2 p.17 L 8-9.

*I think that much of the variability the authors attribute to microorganisms could be attributed to dynamics of melt-out at small scales instead. Inter-annual variations in dark ice extent can be explained by the presence of superimposed ice, perhaps not fully accounted for in MAR. Increases in "Dark Ice Intensity" over time could be related to changes in surface cover within a relatively large MODIS grid box as snow patches and areas of superimposed ice melt away, exposing dark material beneath. The fact that sensible heat flux is a relatively important factor, as is the number of days where temperature is greater than zero suggests that melting of snow and ice could be an important factor independent of biological organisms. Therefore, there appears to be insufficient information to state definitively the cause of the variations in dark ice extent and intensity, although I think the authors have shown that local deposition from year to year can probably be ruled out as a contributing factor.*

*Given a lack of clear evidence supporting a biological source for inter-annual and intra-annual variability in bare ice albedo, I feel that the authors should reduce the emphasis on biological organisms as a source of variability and should also give credence to the possibilities mentioned above.*

*The authors should also address the possibility that the thresholds used here can falsely identify liquid water and possibility even snow or firn as ice or dark ice. The first is probably a minor factor, but the second could potentially lead to a misinterpretation of the results.*

*The work presented here provides a valuable investigation of variations in ice albedo and the presence of impurities in the ablation area of the Greenland ice sheet. I support publication of the study, provided the authors address the points provided in this review.*

We think that there are two distinct issues here. The first concerns the seasonal transition of the surface from being snow-covered, through to firn by metamorphism, possibly a succession to reflective superimposed ice, and finally – assuming enough melting occurs – bare ice. The second concerns why bare ice may be 'dark' (or not) once snow has cleared.

Regarding delineation of snow and superimposed ice from bare/dark ice: as stated in the Methods, we take care to first delineate bare ice from snow-covered surfaces using MODIS band 2 (841-876 nm), which is sensitive to the snow/ice transition via grain size. We therefore do not rely on MAR for this part of the analysis like the comment implies. Only once MODIS observations indicate a pixel is clear of snow do we apply the dark ice threshold to MODIS band 1 (620-670 nm).

Regarding the importance of snow and superimposed ice on dark ice dynamics: it is well known that ablation rates in this sector of the ice sheet are high, on the order of metres per year (e.g. Sole et al., 2013, *GRL*; van As et al., 2016, *GEUS Bulletin*), and so ablating bare ice is usually exposed for much of the melt season. It is therefore hard to envisage a situation in which superimposed ice layers have more than a transient impact upon dark ice intensity, and indeed also to envisage how the small-scale dynamics of snow patches melting could drive regional dark ice dynamics for the entire melt season. Our **addition of the inter-quartile range of bare ice appearance derived from MODIS to Figs 2 and 3** indicates that snow generally clears over a relatively short period near the start of the season (although we acknowledge that sub-pixel patches of snow may still remain).

Furthermore, our observations show that dark ice, as distinct from bare ice is not a static nor omnipresent layer lying beneath snow or superimposed ice. In turn, this suggests that some set of

processes is at work which allows dark ice duration, extent and intensity to vary both through a single melt season after the snow has cleared and between successive melt seasons.

Regarding the comment '*The fact that sensible heat flux is a relatively important factor...suggests that melting of snow and ice could be an important factor independent of biological organisms*', we also examined the relationship between the SHF anomaly (relative to 1981-2010 JJA climatology) and $D_N$ for only the period between snow retreat (as identified by MODIS band 2) and the end of August each year, as opposed to over the entire JJA period. We did not include this analysis in our original manuscript for reasons of space and clarity. The $R^2$ for this relationship is 0.40, almost the same as for JJA SHF (0.41, Figure 5e). We therefore conclude that SHF has an important relationship with bare-ice darkening even after the snow has cleared.

Regarding mixed reflectance: the reflectance threshold we use to delineate dark ice is conservative and field-derived, based on a surface loaded with light algae as compared to a bare ice surface (Appendix A). We apply this threshold – acquired from patches on the order of 20 cm in diameter – to the reflectance value captured for entire MODIS pixels at 600 x 600 m.

The reviewer is correct that there is still potential for mixed reflectance within a MODIS pixel, due for instance to snow patches and superimposed ice, to impact on dark ice intensity values. We already caveat in Sect. 4.3 that we are unable to examine the subpixel extent of dark ice. Considering our field-validated thresholds, the presence of snow/superimposed ice is likely to be more important for pixel-wide dark ice intensity > ~ 0.45, and so we take care in Figure 2 to label the days on which the entire cloud-free extent of the common area has a dark ice intensity < 0.45. For the common area intensity to be < 0.45 but still influenced by high-reflectance snow and/or superimposed ice then there would also need to be widespread 'heavy algae' (Appendix A) in order to pull the area-averaged reflectance down enough to pass the dark ice threshold, which we suggest is unlikely.

More broadly, we have tried to take care to caveat that snow patches and areas of superimposed ice will have an impact on dark ice intensity, especially early in the melt season. As we noted in the manuscript (Sect. 4.3, para. 3), Chandler et al. (2015) reported the presence of a reflective surface immediately after snow clearing, which they attributed to a layer of superimposed ice. But on longer time-scales this is very likely to melt away revealing (as the reviewer notes) 'dark material beneath'. The sensible heat flux is likely to constitute an indirect influence on bare-ice algal assemblages through direct snow (and superimposed ice) removal – which we have already argued in discussion P14, Line 26 onwards.

Following on from our responses above and to the previous general comment, **we have made several small additions to our existing caveats in Sect 4.3, especially para 4 – see manuscript for details.**

*Specific Comments*

*P. 1, Line 1: The recent increases in runoff are not caused by reduced albedo but by changes in atmospheric circulation and atmospheric warming. Albedo changes resulting from these changes amplify melt. Please clarify.*

Thanks for spotting this mistake. Clarified:

Runoff from the Greenland Ice Sheet (GrIS) has increased in recent years due largely to **changes in atmospheric circulation and atmospheric warming. Albedo reductions resulting from these changes have amplified surface melting.**

*P. 1, Line 7: Add "in the future" after "will evolve".*

Done.

*P. 2, Line 6: The statement that "surface melting is controlled by albedo" should be clarified. Other components of the energy balance certainly play a role in controlling melting. Albedo can only play a role with sufficient downward shortwave radiation. Melting can potentially occur during portions of the year when there is less solar radiation as a result of sensible, or longwave fluxes. Please revise this statement, e.g. "Surface albedo plays an important role in modulating surface melt as the surface darkens with warming temperatures…."*

Thanks, we have rephrased following your suggestion:

Surface albedo **plays an important role in modulating the surface melt caused by incoming shortwave radiation.**

*P. 3, Line 34 – P. 4, Line 4: Is there a reference to which the authors can refer here or are these unpublished results of the authors? Please clarify the source in the text.*

Apologies, there was a reference missing here – **ref to Lutz et al (2014) inserted**, who undertook opposed pyranometer measurements.

*P. 4, Line 8: Independent of these processes, there is also the possibility of consolidation of impurities at the surface due to melt, which the authors do discuss later in the manuscript. Perhaps change "inorganic particulate deposition" to "inorganic particulate deposition or redistribution".*

Done.

*P. 4, Lines 28-29: These are all good points, but perhaps now say what the authors think can be done using the thresholds used here.*

Combining the 0.84 and 0.67 um thresholds we can identify bare ice and then distinguish between bare ice that is clean, and bare ice that is significantly darkened by LAI's - which we define as 'dark ice'. This is a significant capability in itself as it enables the mapping of dark ice that needs to be explained. **See also our response to RC1, p.4, line 29, which resulted in changes to this section.**

*P. 4, Lines 29-30: Are the authors saying that some of the variability in extent or intensity could then be associated with grain size evolution and the presence of water? Please clarify.*

Please see our response to RC1, P.4, L29-30, **which resulted in changes to this section.**

*P. 5, Line 1: Clarify how the maximum area was defined, e.g. using daily MODIS reflectance values.*

We already explain how the maximum area is defined in P.5, lines 6-9.

*P. 5, Line 8: Explain why pixels 1 km from the ice sheet margin were removed.*

Pixels near the ice sheet margin can be a mixture of land and ice, and thus are likely to exhibit low reflectance whether or not the ice is dark. To handle locations where the ice mask used here does not precisely match the 600 m resolution of our MODIS data we minimize this 'false positive' by only keeping pixels > 1 km from the margin. We have changed the manuscript as follows:

> Finally, we removed all dark pixels which occurred within ~1 km of the ice sheet margin as **defined by the Greenland Ice Mapping Project (GIMP; Howat et al 2014) in order to remove errant pixels consisting of mixed land and ice cover which remained after applying the GIMP ice area mask.**

*P. 5, Lines 11-12: What is meant by "all the pixels", the number of pixels or fraction of pixels?*

Rephrased:

> First, annual extent ($D_E$) corresponds to the extent **(in km$^2$) covered by the pixels** within the common area which were dark for at least 5 d in each year.

*P. 5, Line 13: Clarify that this is the percentage of all daily cloud-free observations that were classified as "dark" in each JJA period.*

Rephrased:

> Second, annual duration ($D_D$) was defined at each pixel in the common area as the percentage of daily cloud-free observations made in each JJA period **which were classified as dark**, and is thereby normalised for cloud cover.

*P. 5, Line 15-16: It is a bit confusing to refer to this as intensity and to have a lower number indicate a larger intensity. Can't this just be referred to as the average reflectance? Then a lower reflectance is associated with a darker surface.*

Yes it could be in principle. This is a matter of styling preference - we chose $D_I$ in order to correspond with the other dark ice metrics.

*P. 6, Line 6: Include a reference for the ECMWF reanalysis: (Dee et al., 2011) doi:10.1002/qj.828*

Done.

*P. 6, Line 15: Is the daily energy for melt-out "MOF"? Define MOF here. Based on the authors statements it doesn't seem that the MOF is necessarily a proven measure of the conditions needed to produce melt-out. If so it should be made clear that the MOF is suggestive of the conditions needed to cause melt-out, but does not necessarily indicate whether melt-out is occurring or not.*

Yes, the referee is correct that the MOF is not a proven measure of melt-out conditions. We have made the following changes:

> Sect. 2.5: **We therefore characterised the conditions which could cause melt-out of cryoconite holes as the `melt-out flux', MOF**, using... [equation]

> Sect. 3.2.2: We examined the likelihood for cryoconite hole melt-out (causing redistribution of cryoconite materials onto the ice sheet surface) using MOF (Fig. 4c) **which is suggestive of the energy balance conditions that are needed to melt cryoconite holes out of their weathering crust.**

Our response to referee 1 on the topic of MOF is also relevant.

*P. 7, Line 4: Clarify that this "extension is relative to the study of Shimada et al. (2016), which only examined July.*

**Done.**

*P. 7, Line 8: Change "time lag..." to "time lag between tB and the first identified occurrence of dark ice of 10-15 days".*

**Done.**

*P. 7, Line 10: Anticyclonic days don't seem to be shaded gray in Fig. 4.*

We assume the referee means to refer to Fig. 2. Our version of the figure clearly shows the cloudy days in gray; we request that the TC typesetting office confirm this prior to final publication.

*P. 9, Line 6: Change "magnitude of dark ice" to something like "extent and intensity of dark ice" or "extent and reflectivity of dark ice".*

Changed to '**extent and intensity** of dark ice'.

*P. 9, Line 8: Clarify "years when the ice went dark". Perhaps "years when DE was higher" would be more specific.*

We agree – **changed as suggested.**

*P. 9, Line 22: Change "Not only was winter snowfall" to "Not only was 2014-2015 winter snowfall…" for clarity.*

**Changed.**

*P. 11, Line 24: Briefly note how the weathering crust forms.*

Revised, **now notes the importance of subsurface melt by incoming shortwave radiation** (c.f. Cook et al, 2016, Hydro. Proc.).

*P. 12, Line 27: Should "decimeter" be "decameter"?*

Correct, thanks for spotting – **changed.**

*P. 12, Line 21 – P. 13 Line 2: I am not totally convinced by this argument. Much of this could be explained by the presence of superimposed ice, sub-grid scale exposure of bare ice, or even the presence of firn that is mis-classified as bare ice. I don't think the authors can rule out melting as a primary cause of the observed variability, especially since they do not utilize measurements or estimates of melt here. I think the authors should be more careful to acknowledge that melt could be responsible for the observed variability, but that the results also suggest that other factors could be involved.*

Please see our response to the general comment from RC2, above.

*P. 13, Lines 17-25: The variability the authors are discussing seems consistent with the hypothesis of Shimada et al. (2016) except with regard to the changes in dark ice intensity during 2012 and between 2012 and 2013. The statement that "our results reveal a different spatio-temporal pattern" is therefore a bit confusing. As for previous section, the changes in intensity during 2012 could be explained by sub MODIS-grid-scale processes such as melting of snow patches, collecting meltwater. 2012 was a high melt year while 2013 was a low melt year. During 2013, ice is exposed for a much shorter length of time, and the presence of superimposed ice, or again, patches of snow covering the ice could explain the lack of dark ice during that year.*

We note firstly that while stating that our results are 'consistent' with Shimada et al,  this comment does not acknowledge the dark ice dynamics of 2011 and 2012, which was the crux of our discussion and of Shimada et al's argument in favour of cryoconite hole processes. We reiterate that our observations of dark ice dynamics do not support cryoconite hole processes as the source of dark ice variability once the full JJA periods of 2011 and 2012 are taken into account; full details are in the manuscript.

Much of our response to this referee's general comment about *the impact of snow patches and superimposed ice* is relevant to the questioning here of dark ice dynamics during 2012 compared to 2013. In addition, we note that 2012 was the highest melt year on record, and so whilst sub-pixel variability in snow and/or super-imposed ice may have been transiently important to the darkening signal during the start of the melt season in early June, it is highly unlikely that they would have

continued to have an impact on dark ice metrics in July and August. See, for example, Tedstone et al. (2013, *PNAS*), which showed that positive degree days were experienced on almost every single day in this area all the way up to ~1450 m asl until late August.

For 2013, we acknowledge that the melt season was so short that it is possible that snow patches/superimposed ice could have had an impact on dark ice metrics despite MODIS band 2 indicating that the snow had cleared. We now caveat the 2013 statement by saying that prolonged presence of snow patches and/or superimposed ice could have limited dark ice extent:

> This also makes it difficult to explain low $D_E$ in 2013, as cryoconite holes would have needed to form over a short period at the end of summer 2012 in order to sequester cryoconite particles at depth,  unless the presence of snow patches and/or superimposed ice at the surface was so prolonged that only in a few pixels did enough melting take place to expose bare/dark ice.

**We also note that we have added the inter-quartile range of bare ice appearance date to Figures 2 and 3.**

*P. 17, Lines 8-9: The surface must be a mixture of impurities and biological materials, or could even be abiotic. How is the material assumed to be algae?*

The material is indeed a mixture of algal and abiotic impurities; however, microscopic examination showed very clearly that the majority of the impurity load comprised dark coloured algal cells with a relatively very low concentration of mostly clear quartz dust particles. An example of such an example may be found in Yallop et al. (2012, *ISME*). Even by eye, the surface is clearly discoloured mainly by a film of organic matter rather than dust granules which was confirmed to be pigmented algae using a field microscope. **We have added a summary of this information to Appendix A.** Detailed analysis of the constituents will be presented in further papers.

*Figure 1: It would be useful for the reader to include numbers indicating the value of DE for each image.*

**Done.**

*Figure 2: Mention tB in the caption.*

**Done.**

*Figure 3: Note that the snow depth is from MAR. It would be interesting to also see tB in this figure, to allow for a comparison with MAR.*

**We have added ~$t_B$. We have also added the inter-quartile range of the date of bare ice appearance each year as determined from MODIS to both figures 2 and 3.**

*Technical Corrections*

*P. 4, Line 15: Change "cloud" to "clouds".*

**Done.**

*P. 4, Line 28: Change "precisely identify precisely" to "precisely identify"*

**Done.**

*P. 5, Line 18: Add "(DN)" after "normalized darkness" for clarity.*

**Done.**

*P. 5, Line 27: The phrase "with any...only allowed to be cloudy" is confusing. Perhaps just change to "excluding cloudy days".*

Apologies, this interpretation is not correct. Aiming to prevent further confusions we have therefore rephrased as follows:

> Each year, we identified the first rolling window at each pixel that contained at least 3 days of bare or dark ice (not necessarily consecutive) **and 0 days of non-bare or non-dark ice, which therefore permitted up to 4 days of cloud cover in the window.**

*P. 6, Line 12: Place a parenthesis around (T>0) for clarity.*

**Done.**

*P. 7, Line 13: Change "were been" to "were"*

**Done.**

*P. 9, Line 3: Change "not explicable by" to "cannot be explained by"*

**Done.**

*P. 9, Line 14: Change "snowfall which occurs" to "snowfall that occurs"*

**Done.**

[revised manuscript text omitted]